# PSTPIP2 ameliorates aristolochic acid nephropathy by suppressing interleukin-19-mediated neutrophil extracellular trap formation

Changlin Du[†], Chuanting Xu[†], Pengcheng Jia, Na Cai, Zhenming Zhang, Wenna Meng, Lu Chen, Zhongnan Zhou, Qi Wang, Rui Feng, Jun Li, Xiaoming Meng, Cheng Huang*, Taotao Ma*

Inflammation and Immune Mediated Diseases Laboratory of Anhui Province, Anhui Institute of Innovative Drugs, School of Pharmacy, Anhui Medical University, Hefei, China

**\*For correspondence:**
huangcheng@ahmu.edu.cn (CH);
mataotao@ahmu.edu.cn (TM)

[†]These authors contributed equally to this work

**Competing interest:** The authors declare that no competing interests exist.

**Abstract** Aristolochic acid nephropathy (AAN) is a progressive kidney disease caused by herbal medicines. Proline–serine–threonine phosphatase-interacting protein 2 (PSTPIP2) and neutrophil extracellular traps (NETs) play important roles in kidney injury and immune defense, respectively, but the mechanism underlying AAN regulation by PSTPIP2 and NETs remains unclear. We found that renal tubular epithelial cell (RTEC) apoptosis, neutrophil infiltration, inflammatory factor, and NET production were increased in a mouse model of AAN, while PSTPIP2 expression was low. Conditional knock-in of *Pstpip2* in mouse kidneys inhibited cell apoptosis, reduced neutrophil infiltration, suppressed the production of inflammatory factors and NETs, and ameliorated renal dysfunction. Conversely, downregulation of *Pstpip2* expression promoted kidney injury. In vivo, the use of Ly6G-neutralizing antibody to remove neutrophils and peptidyl arginine deiminase 4 (PAD4) inhibitors to prevent NET formation reduced apoptosis, alleviating kidney injury. In vitro, damaged RTECs released interleukin-19 (IL-19) via the PSTPIP2/nuclear factor (NF)-κB pathway and induced NET formation via the IL-20Rβ receptor. Concurrently, NETs promoted apoptosis of damaged RTECs. PSTPIP2 affected NET formation by regulating IL-19 expression via inhibition of NF-κB pathway activation in RTECs, inhibiting RTEC apoptosis, and reducing kidney damage. Our findings indicated that neutrophils and NETs play a key role in AAN and therapeutic targeting of PSTPIP2/NF-κB/IL-19/IL-20Rβ might extend novel strategies to minimize Aristolochic acid I-mediated acute kidney injury and apoptosis.

## Editor's evaluation

This study presents valuable new insights to support Netosis plays an important role in the development of aristolochic acid nephropathy (AAN). A series of compelling experiments using in vivo and in vitro model supported that AAN induced NET formation via IL-19-IL20-β receptor can induce inflammation and cell death. This new knowledge of the interaction between kidney cells and neutrophils could have clinical implications in the treatment of AAN.

## Introduction

The World Health Organization has estimated that 80% of the global population uses traditional medicine (*Kiliś-Pstrusińska and Wiela-Hojeńska, 2021*). However, the pharmacologically active

**eLife digest** Aristolochic acid nephropathy (or AAN for short) is a serious condition affecting the kidneys that is caused by certain traditional Chinese medicines containing a compound called aristolochic acid. This compound is known to have harmful effects on kidney tubular epithelial cells, causing increased inflammation and a form of controlled cell death called apoptosis, which can ultimately lead to organ failure. There is currently no effective treatment for AAN, highlighting the need for a deeper understanding of the mechanisms responsible.

Previous studies have shown that immune cells called neutrophils infiltrate the kidneys and damage cells in the early stages of AAN. Neutrophils produce web-like structures called neutrophil extracellular traps, which have been identified as potentially contributing to the damage. A protein called PSTPIP2, which regulates inflammation, has also been shown to contribute to other types of kidney injury.

To understand how these inflammatory factors might be involved in AAN, Du, Xu et al. genetically engineered mice to produce extra PSTPIP2 protein specifically in their kidneys. When given aristolochic acid, these mice displayed less kidney damage. Further studies of mouse kidney cells showed that PSTPIP2 protects the kidney by suppressing an inflammatory mechanism that leads to the production of neutrophil extracellular traps. By contrast, in models where PSTPIP2 levels were reduced, neutrophil extracellular traps were shown to cause both apoptosis and kidney injury.

The findings of Du, Xu et al. show that neutrophil extracellular traps cause cell damage and apoptosis in a mouse model of AAN and that this action can be reduced by increasing the levels of the protein PSTPIP2. This sheds light on the inflammatory mechanisms behind the kidney damage caused by herbal medicines containing aristolochic acid. Additionally, it opens new avenues for studies aiming to find ways to treat AAN, suggesting that targeting PSTPIP2 could be a promising strategy.

components of some traditional Chinese herbal medicines are associated with adverse effects (*Yang et al., 2018b*). Aristolochic acids (AAs) are found in herbs of the genus *Aristolochia* and are well known for their nephrotoxicity and carcinogenicity (*Bunel et al., 2016*). Notably, 8-methoxy-6-nitro-phenanthro-(3,4-d)–1,3-dioxolo-5-carboxylic acid (AAI) is a major AA (*Jadot et al., 2017*). Aristolochic acid nephropathy (AAN) refers to the deterioration in kidney function associated with toxic AAs (*Anger et al., 2020*). AAN is one of the most serious complications associated with the use of traditional Chinese medicines, with several cases reported globally (*Debelle et al., 2008*). Both mice and rats are susceptible to AA toxicity, allowing in vivo studies on the mechanisms of AA-induced renal toxicity. Acute renal failure with tubular necrosis can be induced in these model animals by a single high dose of AA (*Leong et al., 2021*). Currently, there is no effective treatment for AAN; thus, further investigation is required to elucidate the mechanisms of AAI-induced acute kidney injury (AKI).

Under conditions of persistent injury, signals from the damaged cells activate inflammatory cells, which release proinflammatory cytokines and induce apoptosis of the injured cells. Cell communication is an essential component of mammalian development and homeostasis, ensuring fast and efficient responses to alterations or threats within the host cell (*Jiang et al., 2022a*). Inflammation is the body's protective response against external threats and promotes programmed death of damaged cells to maintain homeostasis, promote the adaptation of tissue to harmful conditions, and restore tissue function (*Medzhitov, 2008*). Several studies have shown that inflammatory cell infiltrates, excessive injury, and death of renal tubular epithelial cells (RTECs) characterize the acute phase of AAN (*Shi et al., 2016*; *Baudoux et al., 2018*; *Jiang et al., 2022b*). Neutrophils, a major inflammatory cell type, infiltrate the kidney in the early stages of AAN, leading to injury. However, it remains unclear how RTEC injury promotes neutrophil activation and how this stimulates apoptosis of damaged RTECs in AAN.

In response to certain stimuli, neutrophils produce neutrophil extracellular traps (NETs)—web-like structures composed of decondensed DNA, histones, and neutrophil granule proteins—through which neutrophils cause tissue damage and interact with other cells. NETs can exacerbate tissue damage via several mechanisms, including promotion of vascular occlusion (*Leppkes et al., 2020*), sterile inflammation (*Huang et al., 2015*), and immune balance disruption. *Saisorn et al., 2021* reported that increased lupus activity may exacerbate AKI through NETs and NET-induced apoptosis. NETs are important for immune defense; however, their role in mediating apoptosis remains controversial. *Lv*

*et al., 2020* reported that NET formation is a major cytotoxic factor causing lung epithelial apoptosis and cytoskeletal destruction. *Shen et al., 2021* also found that NETs promoted trophoblast apoptosis in gestational diabetes mellitus. However, the role of NETs and their potential regulatory mechanisms in AAN remain unclear.

Proline–serine–threonine phosphatase-interacting protein 2 (PSTPIP2), belonging to the Fes/CIP4 homology-bin/amphiphysin/rvs (F-BAR) family, or the pombe cdc15 homology family proteins (*Roberts-Galbraith and Gould, 2010*), is situated on chromosome 18 in both mice and humans (*Gurung et al., 2016*). PSTPIP2 is implicated in immunological and autoinflammatory diseases (*Drobek et al., 2015*) and is expressed not only in various tissues and organs (e.g., heart, liver, lungs) but also in monocytes, mast cells, lymphocytes, and granulocytes (*Zhu et al., 2020*). Studies have demonstrated that PSTPIP2 significantly contributes to inflammatory disorders, tumors, and various diseases by modulating cell proliferation, apoptosis, and the secretion of inflammatory factors (*Chao et al., 2012*; *Grosse et al., 2006*). *Wang et al., 2023* showed that PSTPIP2 is associated with sepsis and can regulate the expression of inflammatory factors by modulating the NF-κB pathway. *Pavliuchenko et al., 2022* reported that molecular interactions involving the adaptor protein PSTPIP2 control neutrophil-mediated responses, leading to autoinflammation (*Pavliuchenko et al., 2022*). Furthermore, in a previous study, we identified a pivotal role of PSTPIP2 in cisplatin-induced AKI (*Zhu et al., 2020*).

In this study, we aimed to determine whether NETs play a role in the survival of injured RTECs and elucidate the underlying mechanism. We provide evidence that RTECs release interleukin (IL)-19 via a PSTPIP2/nuclear factor (NF)-κB-dependent process after AAI stimulation and initiate a feed-forward mechanism involving IL-20Rβ-dependent NET formation. Furthermore, NETs were found to induce apoptosis in AAI-treated RTECs. Overall, tubular epithelial cell-to-neutrophil-to-tubular epithelial cell circulation via NET transfer promoted renal inflammation and apoptosis in AAN. These findings indicate that investigating the regulation of inflammatory factors and the influence of immune cells by PSTPIP2 can promote the understanding of the pathogenesis of AAI-induced AAN. Our elucidation of the interaction between RTECs and neutrophils has potential clinical implications in the prevention and treatment of AAI-induced AAN, providing novel ideas for the identification of new drug targets.

## Results

### PSTPIP2 expression decreased in the AAI-induced acute AAN model in vivo and AAI-treated RTECs in vitro

To investigate the natural course of the renal response to AAI, kidney tissues were harvested on days 1, 3, and 5 after AAI administration (*Figure 1—figure supplement 1A*). Renal impairment, based on elevated serum creatinine (Cr) and blood urea nitrogen (BUN) levels, significantly worsened on days 3 and 5 (*Figure 1—figure supplement 1B and C*). Consistent with the severity of tubular damage observed in hematoxylin and eosin (H&E)-stained sections (*Figure 1A*), levels of the biomarker of tubular damage, KIM-1, markedly increased on days 3 and 5, as detected by western blotting and immunohistochemistry (*Figure 1—figure supplement 1D and E*). The expression of PSTPIP2 in the kidney tissue of the AAI-treated group was significantly lower than that in the vehicle group (*Figure 1B and C*). Immunofluorescence staining of lotus tetragonolobus lectin (LTL) (a proximal tubule marker), calbindin D28k (a distal tubule marker), aquaporin-3 (a collecting duct marker), and PSTPIP2 was performed to explore the location of PSTPIP2 in the setting of AAI-induced AAN or to identify the segments of the nephron with PSTPIP2 expression. As a result, AAI markedly downregulated PSTPIP2 expression in distal tubule epithelial cells and collecting ducts (*Figure 1D*). RTECs were the main site of cell injury during AAI nephrotoxicity. We used AAI to induce AKI in mRTECs in vitro. We evaluated appropriate treatment conditions and found that a 20 hr treatment with AAI did not induce apoptosis in RTECs but increased cell injury; thus, this condition was used for our experiments (*Figure 1—figure supplement 1F–H*). Western blot analysis (*Figure 1E and F*) and IF staining (*Figure 1G and H*) showed that the expression of KIM-1 was significantly upregulated and that of PSTPIP2 was downregulated in AAI-treated mRTECs compared to that in the vehicle-treated group. Collectively, these results indicate that PSTPIP2 expression decreased in both mice and mRTECs treated with AAI.

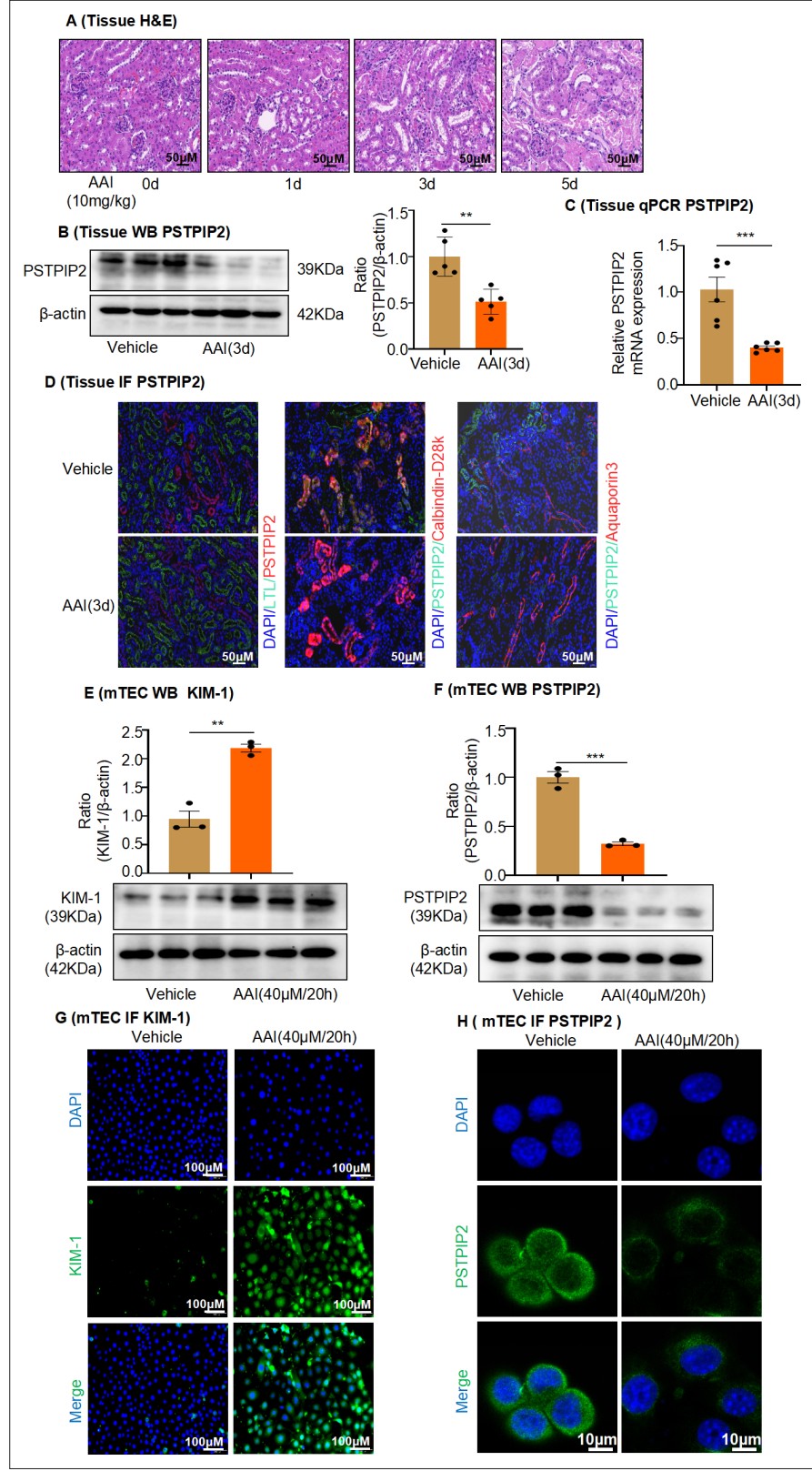

**Figure 1.** PSTPIP2 expression was decreased in the aristolochic acid I (AAI)-induced acute aristolochic acid nephropathy model in vivo and injured mouse renal tubular epithelial cells (mRTECs) in vitro. (**A**) Representative H&E-stained images of kidney sections. Kidneys were isolated on day 0, 1, 3, and 5 after AAI treatment (n = 6 per group). Scale bar, 50 µm. (**B**) Western blotting analysis of PSTPIP2 expression in kidney tissue lysates, which were

*Figure 1 continued on next page*

*Figure 1 continued*

isolated on day 3 after AAI treatment. β-Actin was used as a loading control. Quantification of the PSTPIP2/β-actin ratio (n = 5 per group). (**C**) Real-time PCR analysis of PSTPIP2 expression in kidney tissue, which were isolated on day 3 after AAI treatment. β-Actin was used as a loading control (n = 6 per group). (**D**) Immunofluorescence (IF) staining of PSTPIP2, lotus tetragonolobus lectin (LTL) (a proximal tubule marker), calbindin D28k (a distal tubule marker), and aquaporin-3 (a collecting duct marker) in kidney tissues. DAPI was used for nuclear staining (n = 6 per group). Scale bar, 50 µm. (**E**) Protein levels of kidney injury molecule-1 (KIM-1) in mRTECs treated with AAI (40 µM/20 hr) based on western blotting. Quantification of the KIM-1/β-actin ratio (n = 3). (**F**) Protein levels of PSTPIP2 in mRTECs treated with AAI (40 µM/20 hr) analyzed by western blotting. Quantification of the PSTPIP2/β-actin ratio (n = 3). (**G**) Protein levels of KIM-1 in mRTECs treated with AAI (40 µM/20 hr) analyzed by IF (n = 3). Scale bar, 100 µm. (**H**) Protein levels of PSTPIP2 in mRTECs treated with AAI (40 µM/20 hr) analyzed by IF (n = 3). Scale bar, 10 µm. Data are presented as the mean ± SEM of 3–6 biological replicates per condition. Each dot represents a sample. Significant differences were determined by an independent sample $t$-test (**B, C, E, F**). *$p<0.05$, **$p<0.01$, ***$p<0.001$, ns: nonsignificant.

The online version of this article includes the following source data and figure supplement(s) for figure 1:

**Source data 1.** Data represented by each point in *Figure 1B, C, E, and F*.

**Source data 2.** Uncropped western blots for *Figure 1B, E, and F*.

**Figure supplement 1.** Aristolochic acid I (AAI)-induced acute aristolochic acid nephropathy (AAN) in vivo model and mouse renal tubular epithelial cell (mRTEC) injury in vitro model.

**Figure supplement 1—source data 1.** Data represented by each point in *Figure 1—figure supplement 1B–G*.

**Figure supplement 1—source data 2.** Uncropped western blots for *Figure 1—figure supplement 1E–G*.

## Changes in renal *Pstpip2* expression could affect AAI-induced kidney injury and apoptosis

To determine the role of renal PSTPIP2 levels in AAI-induced AAN, we established a *Pstpip2* conditional knock-in (*Pstpip2*-cKI) mouse model by CRISPR/Cas-mediated genome engineering (*Figure 2A*); PSTPIP2 was overexpressed in the RTECs. *Pstpip2*<sup>Flox/Flox</sup> (FF) mice served as controls. All mice were genotyped using PCR (*Figure 2B*). IF, western blotting, and PCR analyses showed that PSTPIP2 was successfully knocked-in in the mouse kidneys (*Figure 2C–E*). Renal PSTPIP2 overexpression drastically ameliorated renal damage when treated with AAI, based on a histological analysis with H&E staining (*Figure 2F*). Similarly, Cr and BUN levels were remarkably attenuated in AAI-treated *Pstpip2*-cKI mice compared to those in AAI-treated FF mice (*Figure 2G and H*). In addition, western blotting and IHC analysis showed that KIM-1 expression was significantly upregulated in AAI-treated FF mice, while expression in AAI-treated *Pstpip2*-cKI mice was reduced (*Figure 2I and J*).

RTEC apoptosis is a common histopathological feature of AAN. TUNEL staining revealed that TUNEL⁺ cells increased in AAI-treated mice compared to the control mice (*Figure 2—figure supplement 1A*). As caspase-3 plays a central role in the execution phase of apoptosis, we detected its activity and the expression of its activated form, cleaved caspase-3, in mouse kidney tissues. The activity of caspase-3 and level of cleaved caspase-3 were markedly increased in mice treated with AAI (*Figure 2—figure supplement 1B–D*). Conditional knock in of *Pstpip2* significantly decreased TUNEL⁺ cell numbers compared with those in the AAI-treated FF group (*Figure 2—figure supplement 1E*). Furthermore, the activity of caspase-3 and level of cleaved caspase-3 were significantly downregulated in AAI-treated *Pstpip2*-cKI mice compared with those in AAI-treated FF mice (*Figure 2—figure supplement 1F–H*). These results indicate that PSTPIP2 has a protective effect against AAI-induced AAN.

To further examine the functional importance of PSTPIP2, we silenced PSTPIP2 expression using an AAV9-packaged PSTPIP2 shRNA plasmid in *Pstpip2*-cKI mice (*Figure 3A, C–E*). Fluorescence-labeled AAV9–PSTPIP2 was localized to the kidneys (*Figure 3B*). We found that a reduction in PSTPIP2 expression increased serum BUN and Cr levels, along with renal damage (*Figure 3F–H*). Furthermore, immunohistochemical (IHC) staining and western blotting showed that the reduction of PSTPIP2 expression led to the elevated expression of KIM-1 in AAI-induced *Pstpip2*-cKI mice (*Figure 3J and I*). In addition, the reduction of PSTPIP2 expression significantly increased TUNEL⁺ cell numbers, caspase-3 activity, and cleaved caspase-3 levels compared to those in AAI-treated *Pstpip2*-cKI mice (*Figure 3K–N*). These findings indicate that PSTPIP2 plays a key role in AAI-induced renal injury and apoptosis.

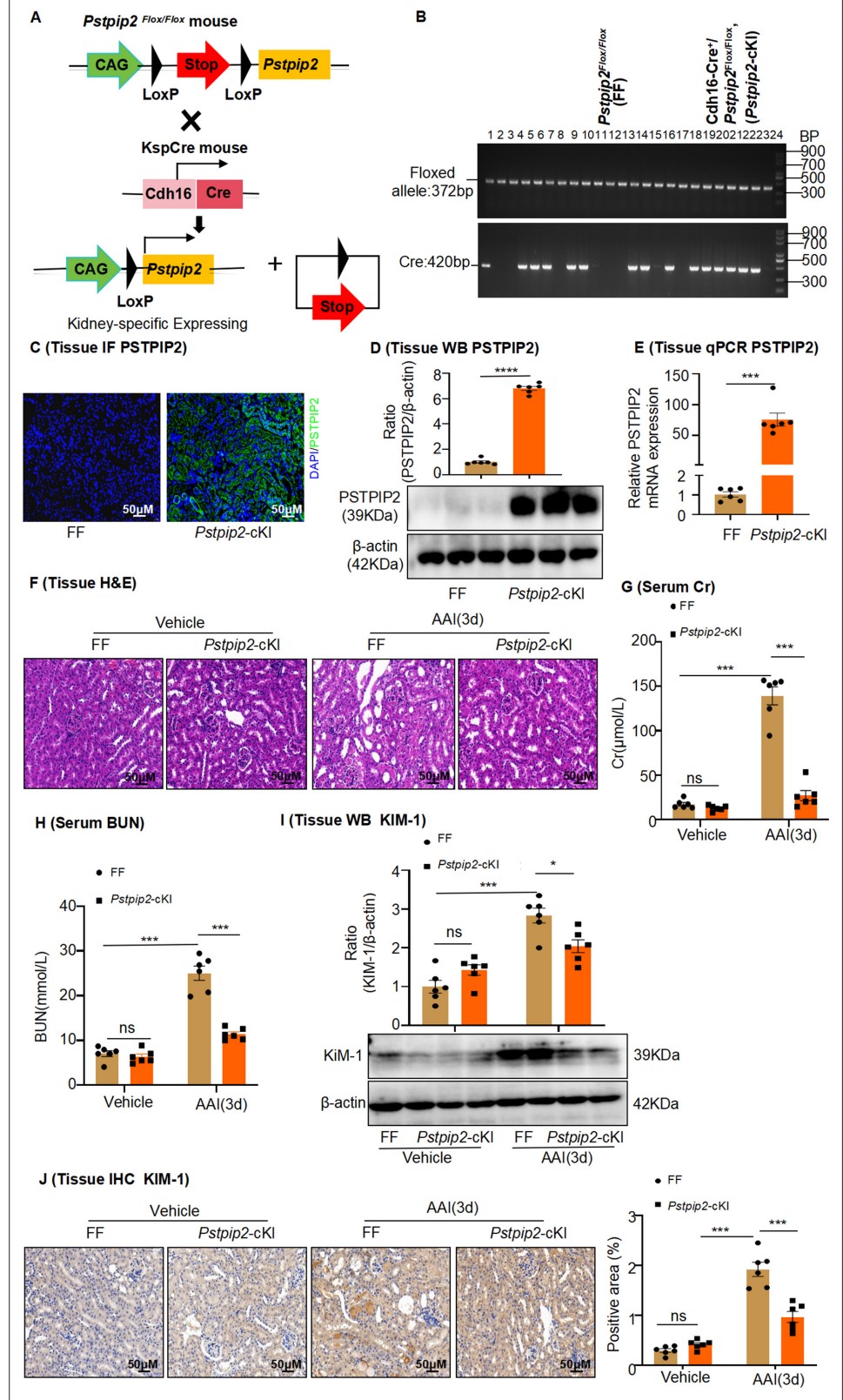

**Figure 2.** *Pstpip2*-cKI attenuates aristolochic acid I (AAI)-induced kidney damage. (**A**) Scheme illustrating the genetic approach for generating *Pstpip2* conditional knock-in (cKI) mice. (**B**) Genotypes of *Pstpip2*-cKI mice were confirmed using PCR. The floxed allele showed a 372-bp-sized band, and the Cre allele was 420 bp; 15 Cdh16-Cre⁺/ *Pstpip2*^Flox/Flox (or *Pstpip2*-cKI) mice (#1, #4–6, #8, #9, #13, #14, #16, #18–23) and 9 *Pstpip2*^Flox/Flox (or FF) mice

*Figure 2 continued on next page*

*Figure 2 continued*

(#2, #3, #7, #10–12, #15, #17, #24) are shown (n = 24). (**C**) Immunofluorescence (IF) staining of *Pstpip2* in FF (n = 6) and *Pstpip2*-cKI (n = 6) mice. Scale bar, 50 μm. (**D**) Protein level of *Pstpip2* in FF (n = 6) and *Pstpip2*-cKI (n = 6) mice assessed by western blotting. (**E**) mRNA level of *Pstpip2* in FF (n = 6) and *Pstpip2*-cKI (n = 6) mice assessed by real-time PCR. (**F**) H&E staining of kidneys from FF (n = 6) and *Pstpip2*-cKI (n = 6) mice treated with AAI. Scale bar, 50 μm. (**G, H**) Serum creatinine (Cr) and blood urea nitrogen (BUN) assay. Serum Cr and BUN showed that *Pstpip2* overexpression prevented renal dysfunction in AAI-induced nephropathy (n = 6 per group). (**I**) Expression of kidney injury molecule-1 (KIM-1) protein in kidney tissues from FF and *Pstpip2*-cKI mice treated with vehicle and AAI examined by western blotting (n = 6 per group). (**J**) Representative immunohistochemical (IHC) staining and KIM-1 levels in kidney tissues (n = 6 per group). Scale bar, 50 μm. Data are presented as the mean ± SEM of six biological replicates per condition. Each dot represents a sample. Significant differences were determined by an independent sample *t*-test (D, E) and one-way ANOVA followed by Tukey's post hoc test (**G**–J). *p<0.05, **p<0.01, ***p<0.001, ns: non-significant.

The online version of this article includes the following source data and figure supplement(s) for figure 2:

**Source data 1.** Data represented by each point in *Figure 2D, E, G, H, I, and J*.

**Source data 2.** Uncropped western blots and gel electrophoresis for *Figure 2B, D, and I*.

**Figure supplement 1.** *Pstpip2*-cKI reduced aristolochic acid I (AAI)-induced kidney apoptosis.

**Figure supplement 1—source data 1.** Data represented by each point in *Figure 2—figure supplement 1A–H*.

**Figure supplement 1—source data 2.** Uncropped western blots for *Figure 2—figure supplement 1C and G*.

## Changes in renal *Pstpip2* expression affect neutrophil infiltration in AAI-induced acute renal injury

Inflammation, characterized by neutrophil infiltration, is concomitant with tubular injury and universally presumed to be implicated in the development of AKI. We detected Ly6G as a neutrophil marker and observed a prominent accumulation of Ly6G+ neutrophils in the kidneys of AAI-treated mice (*Figure 4A*). The infiltrates were accompanied by upregulation of pro-inflammatory TNF-α and chemotactic MCP-1 at the mRNA and protein levels (*Figure 4B–E*). We found that the infiltration of Ly6G+ neutrophils (*Figure 4F*), as well as the mRNA and protein levels of TNF-α and MCP-1 (*Figure 4G–J*), were significantly reduced in AAI-treated *Pstpip2*-cKI mice compared to those in FF mice. IHC staining, ELISA, and real-time PCR analyses showed that downregulation of *Pstpip2* expression induced neutrophil infiltration and inflammatory cytokine production in AAI-induced *Pstpip2*-cKI mice (*Figure 4K–O*). Hence, we concluded that PSTPIP2 may be involved in neutrophil infiltration and cytokine production, which was reduced by renal PSTPIP2 overexpression.

We hypothesized that the transient depletion of neutrophils may provide therapeutic benefits in AAI-induced kidney injury. We examined whether treatment with a specific Ly6G neutralizing antibody 1 d before and after AAI injection could deplete circulating and renal neutrophil levels. Control mice were treated with the same dosage of rat IgG isotype control antibody (*Figure 4—figure supplement 1A*). Compared to the control, we found that the Ly6G antibody could specifically prevent the AAI-induced increase in circulating and renal neutrophil numbers (*Figure 4—figure supplement 1B–D*). Next, we assessed the in vivo therapeutic benefits of neutrophil depletion in AAN. Neutrophil depletion significantly reduced the levels of serum Cr and BUN in AAI-treated mice compared to those in control antibody-treated mice (*Figure 4—figure supplement 2A and B*). In addition, neutrophil depletion in AAI-treated mice significantly attenuated tissue damage and downregulated KIM-1 expression, based on H&E staining and IHC analysis, respectively (*Figure 4—figure supplement 2C and D*), in Ly6G antibody-treated mice. Neutrophil depletion prevented renal dysfunction, highlighting the integral contribution of neutrophils to AAN. Furthermore, we found that neutrophil depletion significantly decreased TUNEL+ cell numbers (*Figure 4—figure supplement 2E*) and significantly downregulated the activity of caspase-3 and level of cleaved caspase-3 following neutrophil depletion in Ly6G antibody-treated mice (*Figure 4—figure supplement 2F and G*). These results suggest that neutrophil depletion ameliorates AAI-induced kidney damage and apoptosis in RTECs.

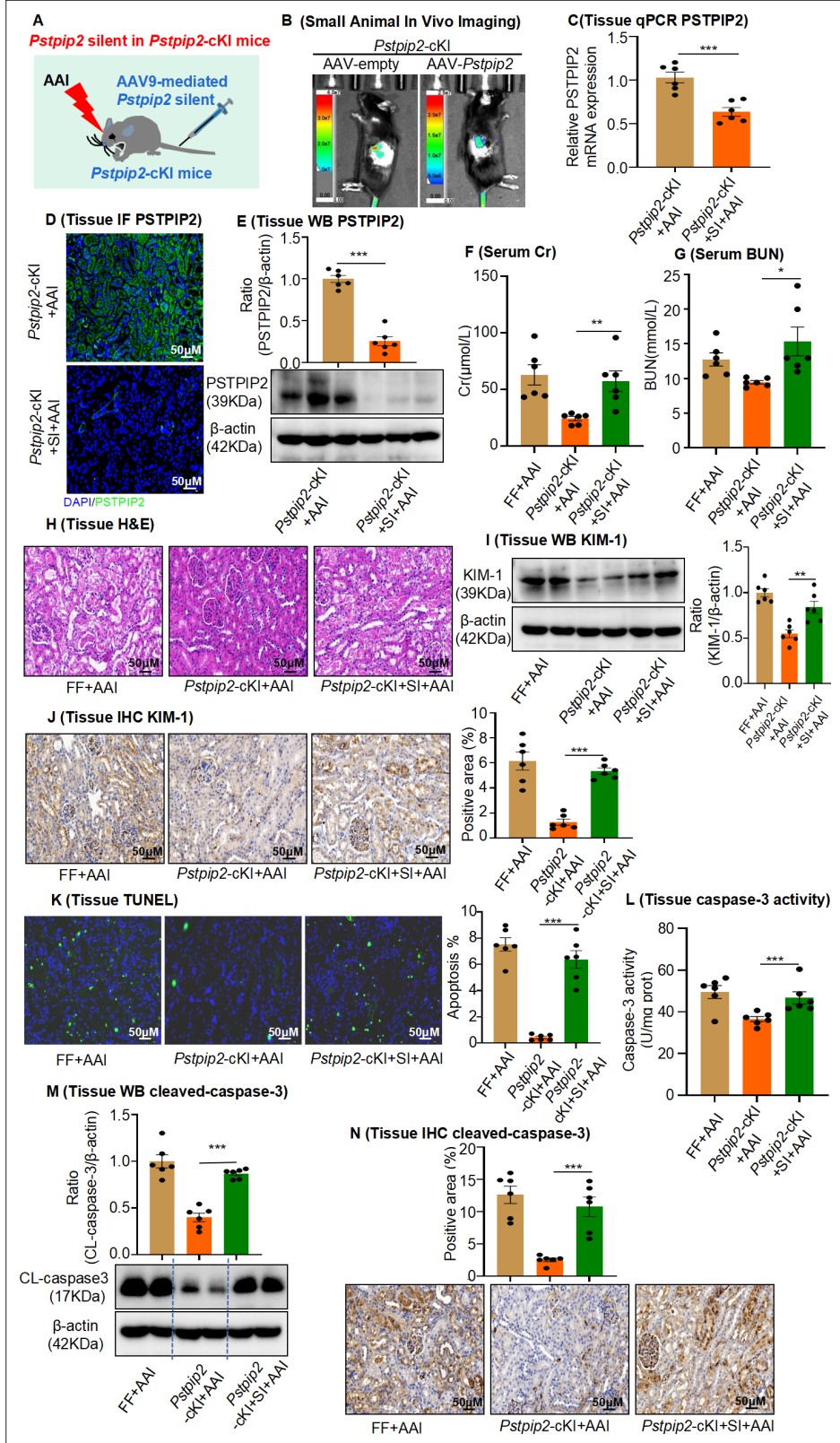

**Figure 3.** AAV9-mediated *Pstpip2* reduction re-induces renal injury and kidney apoptosis in *Pstpip2*-cKI mice. (**A**) Scheme illustrating the rescue experiment in *Pstpip2*-cKI mice. (**B**) Fluorescence-labeled AAV9–PSTPIP2 was localized in the kidney. (**C**) Real-time PCR analysis of PSTPIP2 in kidney tissues (n = 6 per group). (**D**) Immunofluorescence (IF) of PSTPIP2 in kidney tissues (n = 6 per group). Scale bar, 50 μm. (**E**) Western blot analyses

*Figure 3 continued on next page*

*Figure 3 continued*

of PSTPIP2 (n = 6 per group). PSTPIP2 levels were successfully reduced in *Pstpip2*-cKI mice. (**F, G**) Serum creatinine (Cr) and blood urea nitrogen (BUN) assay. Serum Cr and BUN levels showed that silencing PSTPIP2 can aggravate kidney damage in *Pstpip2*-cKI mice treated with aristolochic acid I (AAI) (n = 6 per group). (**H**) Representative H&E-stained images of kidney tissues (n = 6 per group). Scale bar, 50 μm. (**I**) Western blot analysis of KIM-1. Quantification of the KIM-1/β-actin ratio (n = 6). (**J**) Immunohistochemistry (IHC) and kidney injury molecule-1 (KIM-1) levels in kidney tissues (n = 6 per group). Scale bar, 50 μm. (**K**) TUNEL staining and analysis of kidney sections (n = 6 per group). Scale bar, 50 μm. The results showed that silencing PSTPIP2 can aggravate renal apoptosis in *Pstpip2*-cKI mice treated with AAI. (**L**) Detection of caspase-3 activity in kidney tissues (n = 6 per group). (**M**) Western blot analysis of cleaved caspase-3. Quantification of the cleaved caspase-3/β-actin ratio (n = 6). (**N**) IHC and cleaved caspase-3 levels in kidney tissues. Scale bar, 50 μm (n = 6 per group). Data are presented as the mean ± SEM of six biological replicates per condition. Each dot represents a sample. Significant differences were determined by an independent sample *t*-test. *p<0.05, **p<0.01, ***p<0.001, ns: nonsignificant.

The online version of this article includes the following source data for figure 3:

**Source data 1.** Data represented by each point in *Figure 3C, E –G, I–N*.

**Source data 2.** Uncropped western blots for *Figure 3E, I, and M*.

## Changes in *Pstpip2* expression affect NET formation in AAI-induced acute renal injury

Although increased neutrophil infiltration in the kidney is the major cause of AAN, the mechanism of NET formation following AAI exposure remains unknown. To evaluate NET formation, mice received a single IP injection of AAI and were sacrificed after 3 d. Serum nucleosome, MPO, and dsDNA levels were significantly increased after AAI treatment (*Figure 5A–C*). Cit-H3 protein levels, another marker of NETs, were also significantly increased after AAI treatment, as observed by fluorescence microscopy and western blotting (*Figure 5D and E*). We found that serum nucleosome, MPO, and dsDNA levels (*Figure 5F–H*), as well as Cit-H3 levels (*Figure 5I and J*), were significantly lower in AAI-treated *Pstpip2*-cKI mice than those in AAI-treated FF mice. Finally, we found that serum levels of nucleosomes, dsDNA, and MPO, as well as the expression of Cit-H3, were significantly increased in AAI-treated mice after PSTPIP2 expression was reduced compared with those in AAI-treated *Pstpip2*-cKI mice (*Figure 5—figure supplement 1A–E*). Our results demonstrate that PSTPIP2 may play a vital role in the regulation of NET formation.

NET formation during programmed neutrophil cell death (NETosis) has recently been shown to play a pro-injury role in renal inflammatory diseases (*Gupta and Kaplan, 2016*). Accordingly, we explored whether NETosis increases kidney injury induced by AAI. For this purpose, we intraperitoneally injected DNase I or GSK484, a potent inhibitor of NET formation, into the AAI-treated mice (*Figure 5—figure supplement 2A*). Serum nucleosome, MPO, and dsDNA levels (*Figure 5—figure supplement 2B–D*), as well as Cit-H3 levels (*Figure 5—figure supplement 2E and F*), were significantly decreased in mice treated with DNase I or GSK484. We tested the effects of DNase I and GSK484 on the progression of AAN. Treatment with DNase I or GSK484 significantly reduced serum Cr and BUN levels (*Figure 5—figure supplement 3A and B*) and attenuated tissue damage (*Figure 5—figure supplement 3C*) in AAI-treated mice. Moreover, IHC and IF analysis showed that the KIM-1 levels and TUNEL+ cell numbers were significantly decreased in DNase I- or GSK484-treated mice (*Figure 5—figure supplement 3D and E*). The activity of caspase-3 and level of cleaved caspase-3 protein were also significantly downregulated in AAI-treated mice treated with GSK484 or DNase I (*Figure 5—figure supplement 3F and G*). Based on these observations, we surmised that the capacity of neutrophils to cause or exacerbate apoptosis is linked to their NET formation properties. Primary renal neutrophils were isolated from AAI-induced DNase I-treated mice (*Figure 5—figure supplement 4A*), and the levels of dsDNA and neutrophil elastase in their supernatants were examined. We found that dsDNA and neutrophil elastase production could be inhibited by DNase I (*Figure 5—figure supplement 4B and C*). Similarly, Cit-H3 protein levels were significantly decreased in primary renal neutrophils of AAI-induced DNase I-treated mice (*Figure 5—figure supplement 4D and E*). To examine the toxicity of AAI-induced NETs on tubular cells, RTECs were exposed to the supernatant of NETs (*Figure 5—figure supplement 4F*). After 20 hr of incubation, we found decreased numbers of apoptotic cells, as well as downregulated caspase-3 activity and cleaved caspase-3 levels, after DNase I and AAI treatments (*Figure 5—figure*

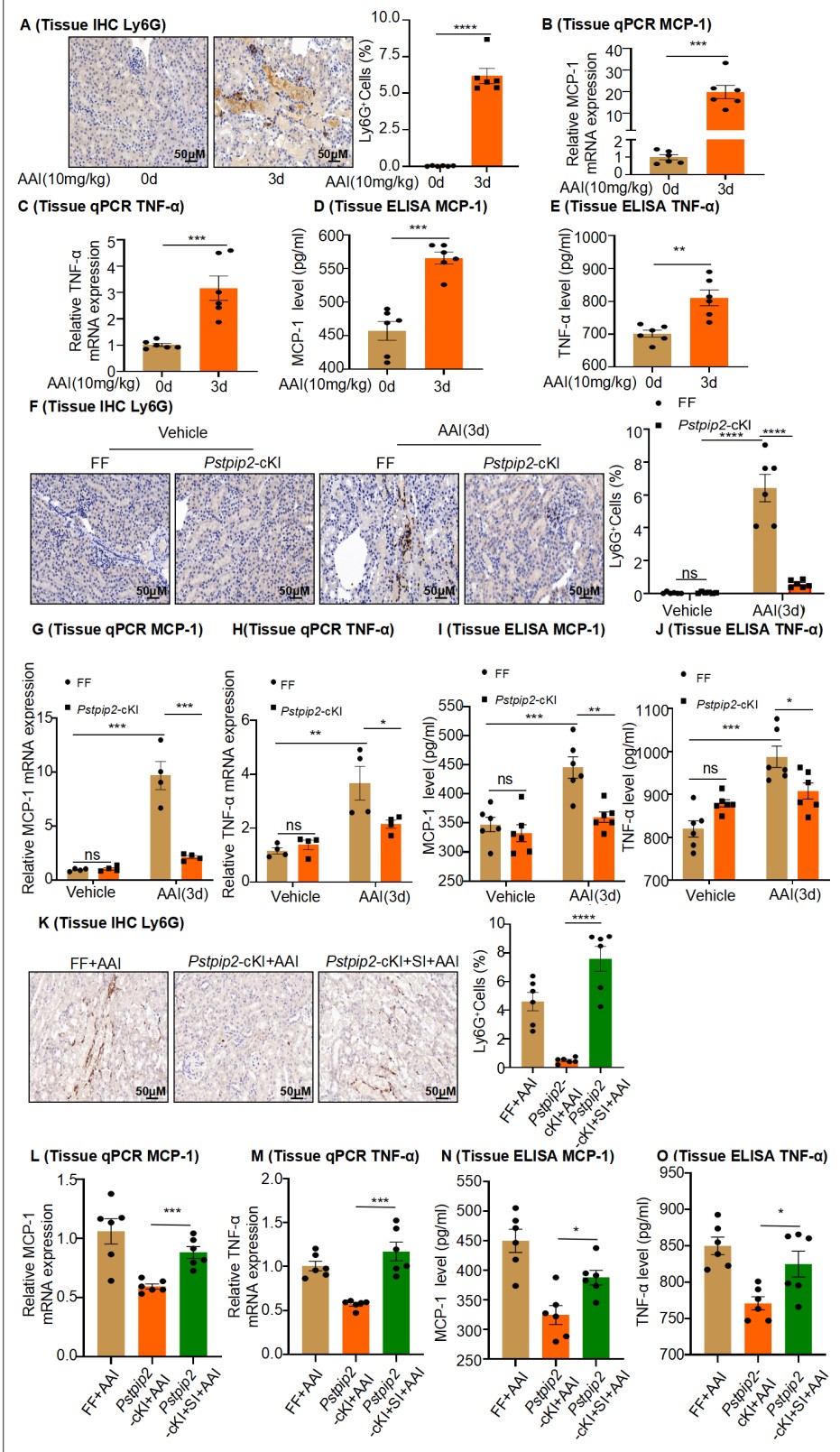

**Figure 4.** Changes in renal *Pstpip2* expression affect neutrophil infiltration and inflammatory factor production in mice with aristolochic acid I (AAI)-induced acute renal injury. (**A**) Representative images of immunohistochemistry (IHC) staining with Ly6G⁺. Scale bar, 50 µm. Quantification of the percentage of Ly6G⁺ neutrophil infiltration. n = 6. (**B, C**) mRNA level of renal tissue monocyte chemoattractant protein-1 (*MCP-1*) and tumor necrosis factor-α

*Figure 4 continued on next page*

*Figure 4 continued*

(*TNF-α*) assessed by real-time PCR (n = 6 per group). (**D, E**) Protein level of renal tissue MCP-1 and TNF-α assessed by ELISA (n = 6 per group). (**F**) Representative IHC staining images of kidney sections, showing the expression of Ly6G. Kidneys were isolated from *Pstpip2*^Flox/Flox and *Pstpip2*-cKI mice on day 3 after AAI treatment. Scale bar, 50 μm. Quantification of the percentage of Ly6G⁺ neutrophil infiltration (n = 6 per group). (**G, H**) Relative mRNA levels of *MCP-1* and *TNF-α* in renal tissue determined by real-time PCR (n = 4 per group). (**I, J**) Protein level of renal tissue MCP-1 and TNF-α assessed by ELISA (n = 6 per group). The results showed that overexpression of PSTPIP2 in the kidneys can reduce renal inflammation in mice treated with AAI. (**K**) Representative IHC staining images of kidney sections, showing the expression of Ly6G. Kidneys were isolated from *Pstpip2*^Flox/Flox, *Pstpip2*-cKI, and *Pstpip2*-cKI-silenced mice on day 3 after AAI treatment. Scale bar, 50 μm. Quantification of the percentage of Ly6G⁺ neutrophil infiltration (n = 6 per group). (**L, M**) Relative mRNA levels of *MCP-1* and *TNF-α* in renal tissue determined by real-time PCR (n = 6 per group). (**N, O**) Protein level of renal tissue MCP-1 and TNF-α assessed by ELISA (n = 6 per group). The results showed that silencing PSTPIP2 can aggravate renal inflammation in *Pstpip2*-cKI mice treated with AAI (n = 6 per group). Data are presented as the mean ± SEM of four to six biological replicates per condition. Each dot represents a sample. Significant differences were determined by an independent sample *t*-test (**B–E** and **K–O**) and one-way ANOVA followed by Tukey's post hoc test (**F–J**). *p<0.05, **p<0.01, ***p<0.001, ns: nonsignificant.

The online version of this article includes the following source data and figure supplement(s) for figure 4:

**Source data 1.** Data represented by each point in *Figure 4A–O*.

**Figure supplement 1.** Mice were pretreated with anti-mouse Ly6G or IgG control antibody followed by aristolochic acid I (AAI) treatment.

**Figure supplement 1—source data 1.** Data represented by each point in *Figure 4—figure supplement 1B–D*.

**Figure supplement 2.** In vivo neutrophil depletion attenuates aristolochic acid I (AAI)-induced kidney damage in mice.

**Figure supplement 2—source data 1.** Data represented by each point in *Figure 4—figure supplement 2A, B, and D–G*.

*supplement 4G–J*). These results suggest that inhibition of NET formation attenuates AAI-induced kidney damage.

## PSTPIP2 inhibits NF-κB signaling to reduce the release of IL-19 from damaged RTECs, thereby reducing neutrophil infiltration and NET formation

To better characterize the specific role of PSTPIP2 in RTECs, overexpression of PSTPIP2 was induced by transfection with the PSTPIP2 plasmid (pEGFP-C1- PSTPIP2) in mRTECs. We confirmed that the pEGFP-C1- PSTPIP2 plasmid elevated PSTPIP2 expression (*Figure 6—figure supplement 1D and E*). We stimulated RTECs with AAI for 20 hr to explore whether they secreted important cytokines (*Figure 6A*). Real-time PCR analysis showed that the expression of IL-19 was most significantly elevated compared with that of other interleukins in AAI-treated mRTECs, and overexpression of PSTPIP2 significantly decreased IL-19 expression compared with that in control plasmid-transfected cells (*Figure 6B*). We found that the level of IL-19 was significantly increased in the supernatant of AAI-treated mRTECs (*Figure 6C*), consistent with a marked increase in its mRNA and protein levels (*Figure 6D and E*). Conversely, we found that the expression of IL-19 was significantly reduced after overexpression of PSTPIP2 in mRTECs (*Figure 6K–L*, *Figure 6—figure supplement 1F*). Using immunofluorescence staining, we observed that E-cadherin was colocalized with IL-19, suggesting that RTECs are the major source of IL-19 after AAI-induced renal injury (*Figure 6—figure supplement 1A*). Increased serum IL-19 levels have been observed in various kidney diseases, including in various kidney diseases, including hydronephrosis accompanied by ureteral stones, diabetes, and polycystic kidney disease, as well as in kidney transplant recipients (Table 2). Notably, serum IL-19 and Cr levels were positively correlated ($R = 0.5472$, p<0.05; *Figure 6F*). The expression of IL-19 was significantly lower in AAI-treated *Pstpip2*-cKI mice than that in AAI-treated FF mice (*Figure 6G and I*, *Figure 6—figure supplement 1B*). However, the downregulation of PSTPIP2 expression significantly increased IL-19 expression after stimulation with AAI (*Figure 6H and J* and *Figure 6—figure supplement 1C*).

To confirm whether IL-19 can accelerate AAN and to identify the functional role of IL-19 in NET formation in mice intraperitoneally injected with AAI, recombinant mouse IL-19 (rIL-19) was

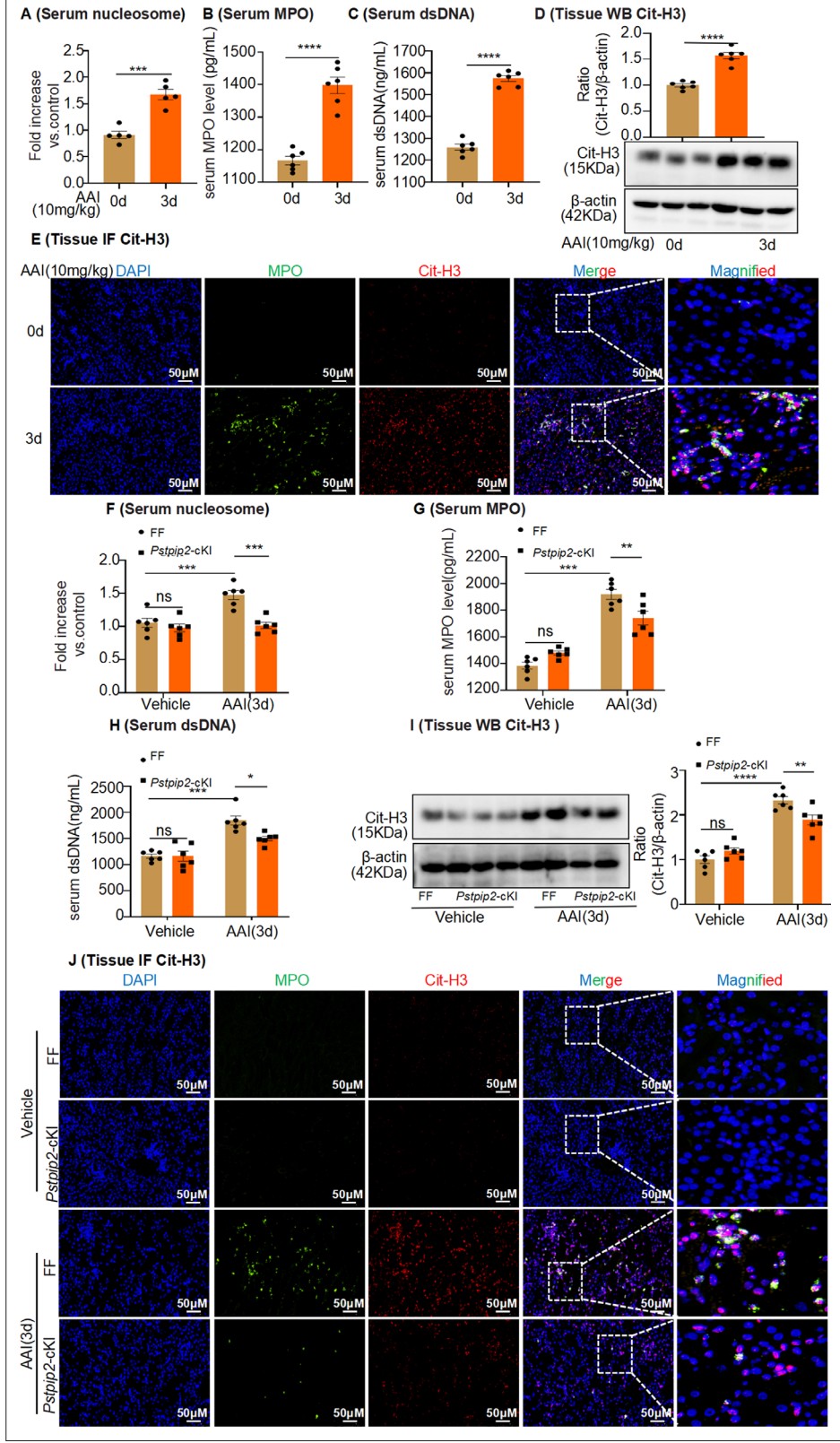

**Figure 5.** *Pstpip2* conditional knock-in in the kidney significantly decreased neutrophil extracellular trap (NET) formation. (**A**) Serum nucleosome level after aristolochic acid I (AAI) treatment for 3 d (n = 5). (**B, C**) Serum myeloperoxidase (MPO) and dsDNA level after AAI treatment for 3 d (n = 6). (**D**) Protein level of citrullinated histone 3 (Cit-H3) assessed by western blotting after AAI treatment for 3 d (n = 6). (**E**) Immunofluorescent (IF)

*Figure 5 continued on next page*

*Figure 5 continued*

staining revealed increased co-localization of MPO (green) and Cit-H3 (red) after AAI treatment for 3 d (n = 6). Scale bar, 50 µm. (**F**) Serum nucleosome level of *Pstpip2*[Flox/Flox] and *Pstpip2*-cKI mice treated with AAI (n = 6 per group). (**G, H**) Serum MPO and dsDNA level of *Pstpip2*[Flox/Flox] and *Pstpip2*-cKI mice treated with AAI (n = 6 per group). (**I**) Western blot analysis of Cit-H3. Quantification of the Cit-H3/β-actin ratio (n = 6 per group). (**J**) Representative IF staining images of kidney sections, showing the expression of Cit-H3 (red) and MPO (green). Kidneys were isolated from *Pstpip2*[Flox/Flox] and *Pstpip2*-cKI mice on day 3 after AAI treatment (n = 6 per group). Scale bar, 50 µm. Data are presented as the mean ± SEM of six biological replicates per condition. Each dot represents a sample. Significant differences were determined by an independent sample *t*-test (**A–D**) and one-way ANOVA followed by Tukey's post hoc test (**F–I**). *$p<0.05$, **$p<0.01$, ***$p<0.001$, ****$p<0.0001$, ns: nonsignificant.

The online version of this article includes the following source data and figure supplement(s) for figure 5:

**Source data 1.** Data represented by each point in *Figure 5A–D and F–I*.

**Source data 2.** Uncropped western blots for *Figure 5D and I*.

**Figure supplement 1.** AAV9-mediated *Pstpip2* restoration re-induces neutrophil extracellular trap (NET) formation in *Pstpip2*-cKI mice.

**Figure supplement 1—source data 1.** Data represented by each point in *Figure 5—figure supplement 1A–D*.

**Figure supplement 1—source data 2.** Uncropped western blots for *Figure 5—figure supplement 1D*.

**Figure supplement 2.** Neutrophil extracellular trap (NET) formation inhibitor, GSK484, or DNase I inhibited formation of NETs in aristolochic acid I (AAI)-treated mice.

**Figure supplement 2—source data 1.** Data represented by each point in *Figure 5—figure supplement 2B–E*.

**Figure supplement 2—source data 2.** Uncropped western blots for *Figure 5—figure supplement 2E*.

**Figure supplement 3.** Inhibition of neutrophil extracellular traps (NETs) formation attenuates aristolochic acid I (AAI)-induced kidney damage and apoptosis.

**Figure supplement 3—source data 1.** Data represented by each point in *Figure 5—figure supplement 3A, B, and D-G*.

**Figure supplement 4.** Conditional medium (CM) derived from primary neutrophils of aristolochic acid nephropathy (AAN) mice co-treated with neutrophil extracellular traps (NETs) inhibitor prevents apoptosis of mouse renal tubular epithelial cells (mRTECs).

**Figure supplement 4—source data 1.** Data represented by each point in *Figure 5—figure supplement 4B, C, E, and H–J*.

**Figure supplement 4—source data 2.** Uncropped western blots for *Figure 5—figure supplement 4E and J*.

administered to mice immediately after treatment with AAI. Histological analysis with H&E staining revealed that intraperitoneal injection of rIL-19 at a dose of 4 µg caused no damage in vehicle-treated mice but considerably accelerated renal damage when combined with AAI treatment (*Figure 7A*). Similarly, intraperitoneal injection of IL-19 significantly increased the levels of Cr and BUN in the serum of AAI-treated mice (*Figure 7B and C*). In addition, western blotting and PCR and IHC analysis showed that KIM-1 expression was significantly upregulated in IL-19-treated AAN mice compared with vehicle-treated AAN mice (*Figure 7D–F*). Furthermore, we found that rIL-19 significantly increased TUNEL[+] cell numbers (*Figure 7I*) and significantly upregulated the activity of caspase-3 and level of cleaved caspase-3 following intraperitoneal injection of IL-19 in AAI-treated mice (*Figure 7G and H*). These results suggest that injection of IL-19 protein accelerates AAN. Intraperitoneal injection with AAI resulted in increased numbers of MPO[+] neutrophils (*Figure 8C*) in the injured kidney, as well as increased kidney tissue levels of TNF-α and MCP-1 in rIL-19-treated mice compared to vehicle-treated mice. (*Figure 8A and B*). Similarly, NET formation, as measured by serum levels of MPO, dsDNA, and nucleosomes, as well as by tissue levels of Cit-H3 in the injured kidney, significantly increased in rIL-19-treated AAN mice compared to vehicle-treated AAN mice (*Figure 8D–H*). These results also indicated that IL-19 could further drive the inflammatory response and NET formation in AAN.

IL-19 is a potent cytokine in neutropenia treatment (*Xiao et al., 2021*). We assessed whether IL-19 is chemotactic to mouse bone marrow-derived neutrophils using a Transwell system (*Figure 9A*) and found that IL-19 stimulation was able to induce neutrophil migration, particularly at a concentration of 80 ng/mL (*Figure 9B*). Although IL-19 has been shown to enhance neutrophil recruitment in AAN mice (*Figure 8C*), it is unclear whether IL-19 induces NET formation. We used a series of different concentrations of recombinant IL-19 (rIL-19) to stimulate mouse bone marrow-derived neutrophils

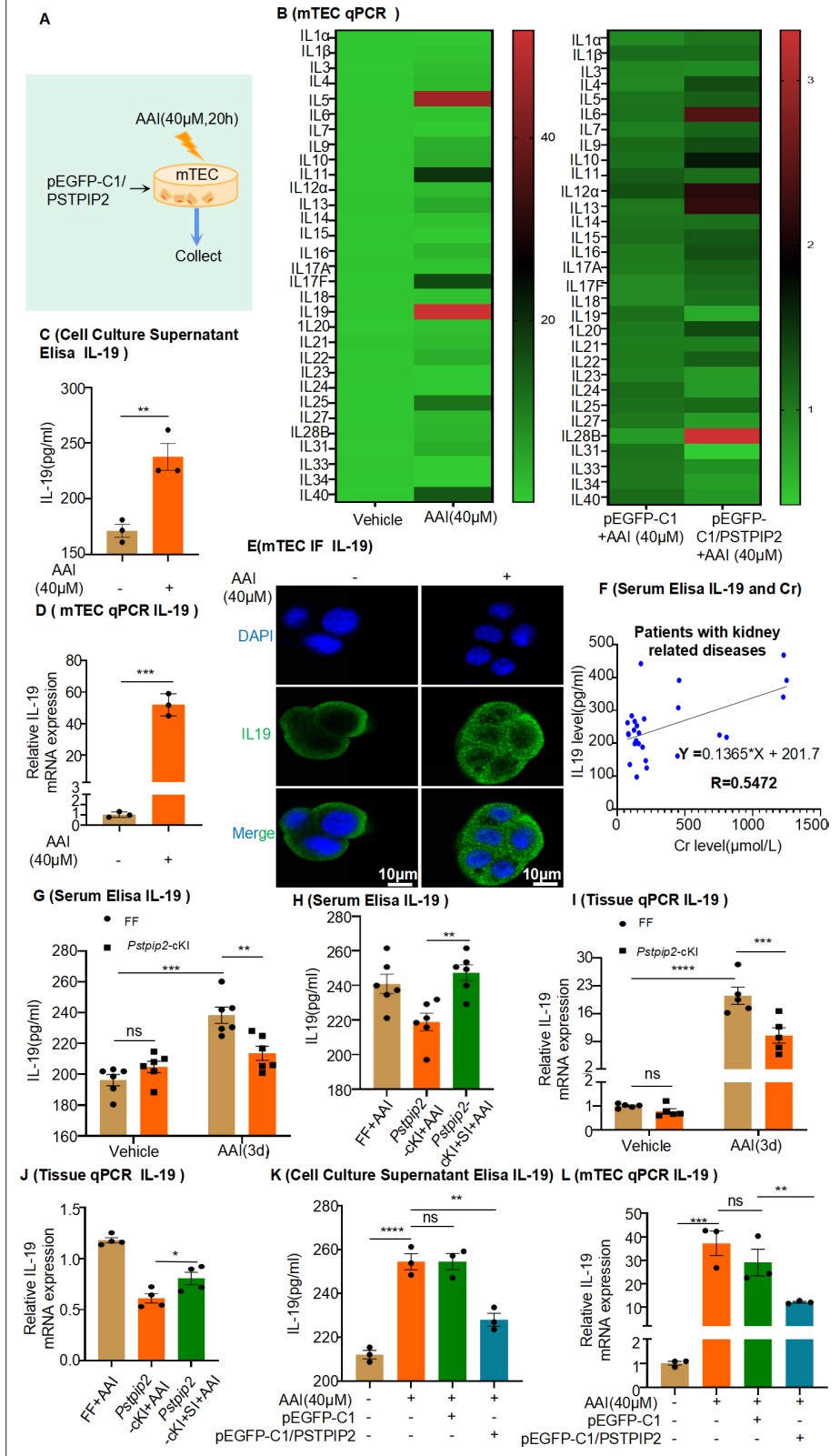

**Figure 6.** Expression of interleukin-19 (IL-19) was associated with kidney injury and regulated by PSTPIP2. (**A**) PSTPIP2 was overexpressed in mouse renal tubular epithelial cells (mRTECs) by the plasmid pEGFP-C1/PSTPIP2; the cells were subsequently treated with aristolochic acid I (AAI, 40 µM) for 20 hr. (**B**) mRNA levels of various interleukins in mRTECs assessed by real-time PCR (n = 3). (**C**) Protein level of IL-19 in the cell culture supernatant

*Figure 6 continued on next page*

*Figure 6 continued*

assessed by ELISA (n = 3). (**D**) mRNA level of IL-19 in mRTECs assessed after treatment with AAI by real-time PCR (n = 3). (**E**) IL-19 levels were assessed by immunofluorescence (IF) staining in cells treated with AAI. Scale bar, 10 μm (n = 3). (**F**) Correlation analysis of serum IL-19 and Cr levels in patients with kidney-related diseases (n = 27). (**G**) Serum levels of IL-19 protein in *Pstpip2*^Flox/Flox^ and *Pstpip2*-cKI mice treated with AAI (n = 6). (**H**) Serum levels of IL-19 protein in *Pstpip2*^Flox/Flox^, *Pstpip2*-cKI, and *Pstpip2*-cKI-silenced mice on day 3 after AAI treatment (n = 6). (**I**) Relative mRNA levels of IL-19 in *Pstpip2*^Flox/Flox^ and *Pstpip2*-cKI mice treated with AAI determined by real-time PCR (n = 5 per group). (**J**) Real-time PCR analysis of IL-19 (n = 4 per group). (**K**) Protein level of IL-19 in cell culture supernatant assessed by ELISA (n = 3). (**L**) mRNA level of IL-19 in mRTECs assessed by real-time PCR (n = 3). Data are presented as the mean ± SEM of 3–6 biological replicates per condition. Each dot represents a sample. Significant differences were determined by an independent sample *t*-test (**B–D, H, J**) and one-way ANOVA followed by Tukey's post hoc test (**G, I, K, L**). Data in (**F**) were analyzed by Pearson correlation.*p<0.05, **p<0.01, ***p<0.001, ****p<0.0001, ns: non-significant.

The online version of this article includes the following source data and figure supplement(s) for figure 6:

**Source data 1.** Data represented by each point in *Figure 6B–D and F–L*.

**Figure supplement 1.** Expression of interleukin-19 (IL-19) in mouse kidneys and mouse tubular epithelial cells (mRTECs) in aristolochic acid I (AAI)-induced aristolochic acid nephropathy (AAN).

**Figure supplement 1—source data 1.** Data represented by each point in *Figure 6—figure supplement 1D and E*.

**Figure supplement 1—source data 2.** Uncropped western blots for *Figure 6—figure supplement 1E*.

---

(*Figure 6—figure supplement 1G*) for 4 hr (*Figure 9C*). The protein levels of IL-20Rβ, the IL-19 membrane receptor, increased considerably after stimulation with various rIL-19 concentrations (*Figure 9G and I*), suggesting that neutrophils express IL-20Rβ, which is upregulated in response to rIL-19 exposure. We sought to determine whether IL-19 induces NET formation. Accordingly, we measured nucleosome, MPO, and dsDNA levels in neutrophil media after stimulation with rIL-19 and found a significant increase in all levels at a concentration of 80 ng/mL compared to those in the vehicle group (*Figure 9D–F*). Cit-H3 protein levels were significantly increased in IL-19-treated mouse bone marrow-derived neutrophils (*Figure 9G and H*). Under similar conditions, the neutrophils lost their nuclei and released strings and cellular debris after IL-19 treatment, as observed with SEM (*Figure 9J*). To examine the toxicity of IL-19-induced NETs on damaged tubular cells, RTECs were exposed to the supernatant of NETs (*Figure 9—figure supplement 1A*); at 20 hr of incubation, we found an increased number of apoptotic RTECs (*Figure 9—figure supplement 1D and E*). Consistently, the activity of caspase-3 and level of cleaved caspase-3 protein were significantly upregulated after stimulation with IL-19-derived NETs (*Figure 9—figure supplement 1B and C*).

The transcription factor NF-κB has been reported to regulate IL-19 gene transcription (*Xiao et al., 2021*), but it is unclear whether PSTPIP2 regulates IL-19 expression by activating or inhibiting the NF-κB pathway. In addition, to verify the correlation between PSTPIP2 and NF-κB p65, we used mRTEC extract for co-immunoprecipitation (Co-IP) experiments using anti-NF-κB p65 and anti-PSTPIP2 antibodies. The Co-IP results confirmed an interaction between PSTPIP2 and NF-κB p65 in mRTECs (*Figure 10A*). As expected, NF-κB was strongly activated in AAI-treated mRTECs, as evidenced by the markedly increased phosphorylation (S536) and nuclear localization of NF-κB p65, accompanied by enhanced phosphorylation of IκBα (S32/36), and overexpression of PSTPIP2 inhibited AAI-induced activation of NF-κB signaling (*Figure 10B and C*). In addition, IF staining showed that the overexpression of PSTPIP2 significantly inhibited the nuclear transfer of p65 (*Figure 10D*). The NF-κB pathway was inhibited after a 2 hr pretreatment with the NF-κB inhibitor PDTC (25 μM; *Figure 10F*), immediately after which, we removed the cell supernatant and retreated the cells with AAI for 20 hr (*Figure 10E*). Real-time PCR analysis showed that IL-19 was visibly downregulated in the PDTC + AAI-treated group compared to the AAI-treated group (*Figure 10G*). Collectively, these results indicate that PSTPIP2 inhibits NF-κB signaling to reduce IL-19 release from the RTECs.

## Discussion

This study provides a new perspective on the role of PSTPIP2 in acute AAN and identifies a potential mechanism of PSTPIP2 in inhibiting epithelial apoptosis to alleviate AAI-induced nephrotoxicity.

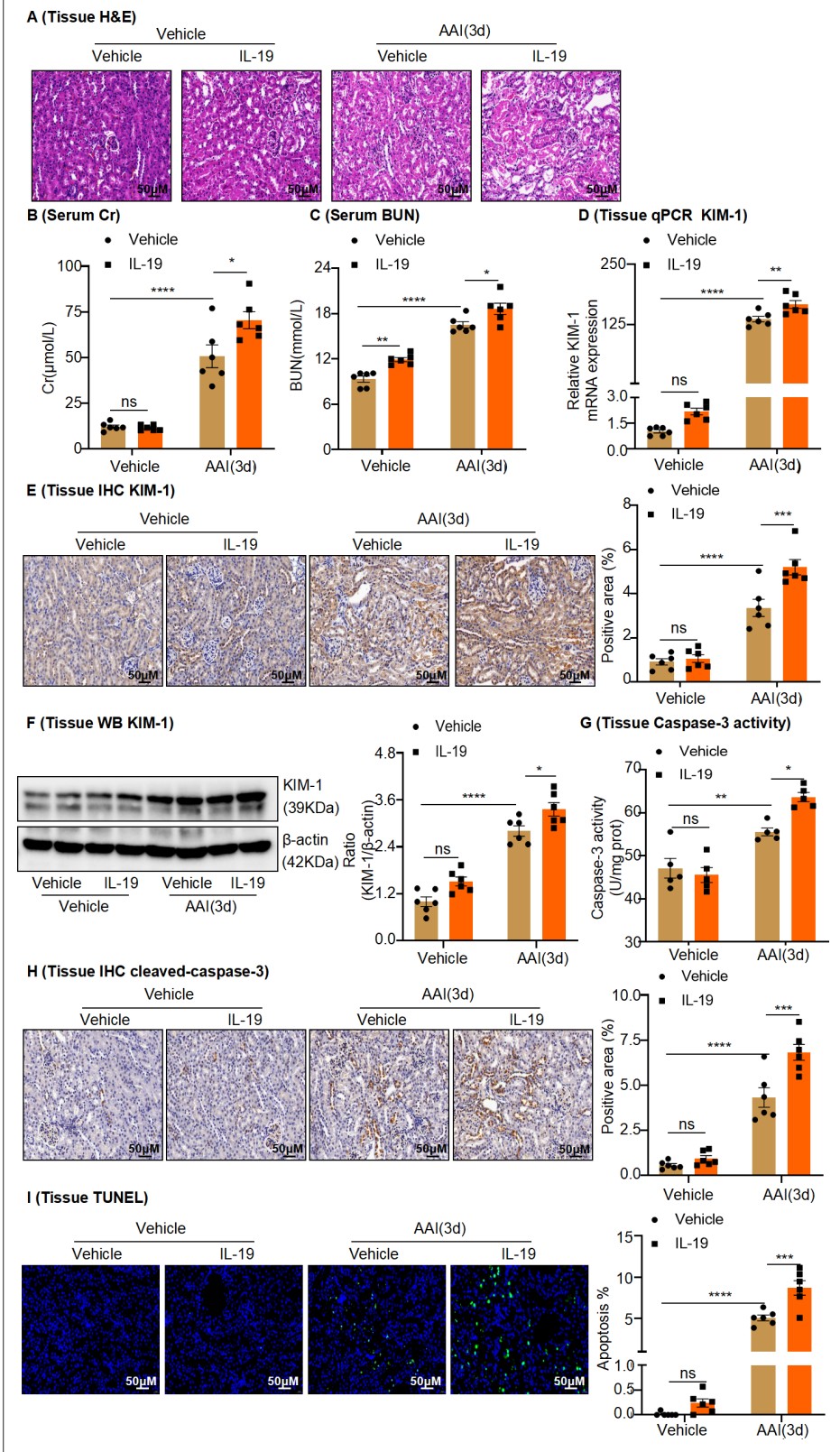

**Figure 7.** In vivo administration of rIL-19 in mice exacerbated the progression of aristolochic acid nephropathy. (**A**) Representative H&E staining images of kidney sections from vehicle and rIL-19-treated mice after being administered with aristolochic acid I (AAI) (n = 6 per group). Scale bar, 50 μm. (**B, C**) Serum creatinine (Cr) and blood urea nitrogen (BUN) levels assessed after treatment with AAI and co-treatment with vehicle or rIL-

*Figure 7 continued on next page*

*Figure 7 continued*

19 (n = 6 per group). (**D**) mRNA level of renal tissue KIM-1 assessed by real-time PCR (n = 6 per group). (**E**) Immunohistochemistry (IHC) staining and quantification of kidney injury molecule-1 (KIM-1) in kidney tissues (n = 6 per group). Scale bar, 50 µm. (**F**) Protein level of renal tissue KIM-1 assessed by western blotting (n = 6 per group). (**G**) Detection of caspase-3 activity in kidney tissues (n = 5 per group). (**H**) Immunohistochemistry (IHC) staining and quantification of cleaved-caspase-3 in kidney tissues (n = 6 per group). Scale bar, 50 µm. (**I**) TUNEL staining and quantification in kidney sections (n = 6 per group). Scale bar, 50 µm. Data are presented as the mean ± SEM of 5–6 biological replicates per condition. Each dot represents a sample. Significant differences were determined by one-way ANOVA followed by Tukey's post hoc test (**B–I**). *p<0.05, **p<0.01, ***p<0.001, ns: nonsignificant.

The online version of this article includes the following source data for figure 7:

**Source data 1.** Data represented by each point in *Figure 7B–I*.

**Source data 2.** Uncropped western blots for *Figure 7F*.

We confirmed that PSTPIP2 was significantly downregulated in both AAI-induced acute AAN mouse models and AAI-exposed mRTECs. We also found that the conditional knock-in of *Pstpip2* in mouse kidneys was sufficient to ameliorate renal dysfunction, histological injury, cell apoptosis, and inflammatory responses induced by AAI in acute AAN mouse models. Accordingly, silencing PSTPIP2 in the kidney of *Pstpip2*-cKI mice promoted renal injury and cell apoptosis, confirming the critical role of PSTPIP2 in acute AAN. Mechanistically, we found that PSTPIP2 served as a negative regulator of acute AAN, at least in part, by reducing apoptosis of RTECs by regulating the IL-19-mediated formation of NETs. Our results, therefore, support the potential of PSTPIP2-targeted therapy in patients with acute AAN.

First, we examined the role of PSTPIP2 in renal injury and inflammation. PSTPIP2 plays a crucial role in autoimmune diseases and is involved in macrophage activation, neutrophil migration, and cytokine production (*Xu et al., 2022*). Studies have shown that the protein tyrosine phosphatase PTP-PEST, Src homology domain-containing inositol 5'-phosphatase 1 (SHIP1), and C-terminal Src kinase (CSK) can bind to PSTPIP2 and inhibit the development of autoinflammatory diseases (*Xu et al., 2021*). *Yang et al., 2018a* demonstrated that PSTPIP2 could ameliorate the degree of liver fibrosis and hepatic inflammation in CCl$_4$-induced hepatic fibrosis. Neutrophils represent the first line of innate immune defense to clear or contain invading pathogens. In AAN, which is characterized by increased inflammation and cell death (*Li et al., 2021*), an increase in renal neutrophil levels was associated with poor clinical outcomes (*Wang et al., 2021*). Recent studies have also shown that increased neutrophil levels can exacerbate AKI (*Cho et al., 2017*; *Bolisetty and Agarwal, 2009*). We observed a large infiltration of neutrophils in the kidneys and a significant increase in the expression of pro-inflammatory TNF-α and chemotactic MCP-1 molecules 3 d after establishing acute AAN. However, conditional knock-in of *Pstpip2* in the kidney markedly reduced neutrophil infiltration and the expression of inflammatory factors, both of which were significantly increased after reducing PSTPIP2 levels in the kidney. Given the important role of neutrophils in the pathological mechanism underlying AKI, we assessed whether transient neutrophil depletion would induce similar therapeutic benefits in AAN. We found that a Ly6G-specific monoclonal antibody (*Bukong et al., 2018*) could deplete both circulating and renal neutrophil levels in mice, reducing systemic inflammation and renal injury. Considering that neutrophils are an important immune cell type, transient neutrophil depletion may be feasible in the treatment of AAN.

Over the past decade, many new theories regarding the pathogenesis of AAN have emerged (*Chan and Ham, 2021*; *Kocic et al., 2021*; *Jin et al., 2020*). However, the pathophysiological mechanisms by which AAI induces renal injury remain largely unknown. Neutrophils are rapidly recruited to tissues during infection to eliminate pathogens, both by phagocytosis and the formation of NETs in the extracellular space (*Amini et al., 2018*). NETs are one of the most widely studied pathways of neutrophil-induced injury and have been observed in kidney injury induced by various stimuli (*Wang et al., 2022*). Thus, there is a growing interest in the synthesis and regulation of NETs. NET formation is usually associated with neutrophil death, or NETosis, which is morphologically distinct from apoptosis and necrosis (*Zawrotniak and Rapala-Kozik, 2013*). NET formation depends on the activation of peptidyl arginine deiminase (PAD) enzymes that convert arginine residues of histones to citrulline (*Raup-Konsavage et al., 2018*). Histone citrullination neutralizes DNA–histone interactions, resulting in chromatin decondensation and NET release (*Nakazawa et al., 2017*). NETs are composed of

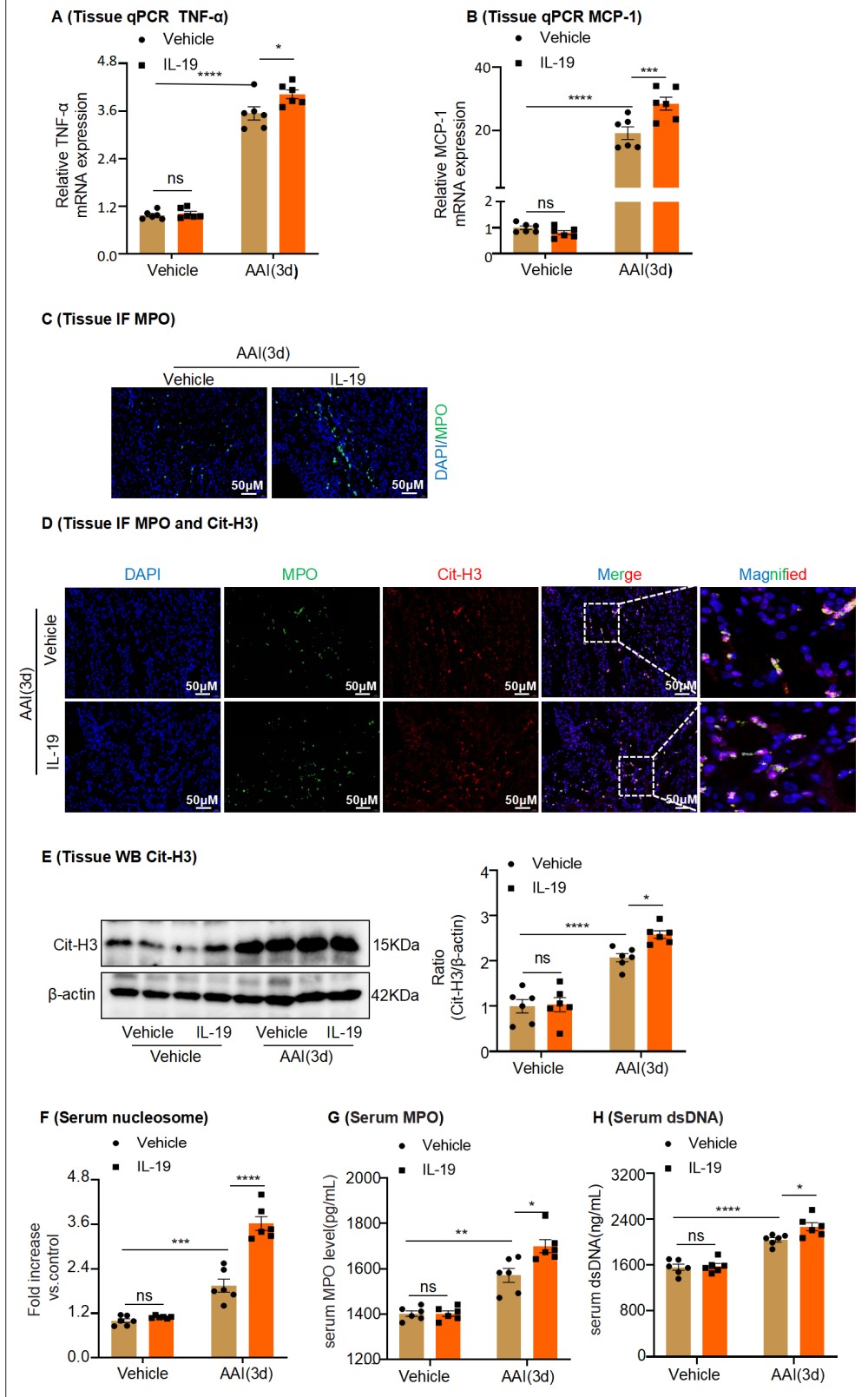

**Figure 8.** Administration of rIL-19 in mice promotes neutrophil infiltration, inflammatory factor production, and formation of neutrophil extracellular traps in aristolochic acid nephropathy. (**A, B**) mRNA level of renal tissue monocyte chemoattractant protein-1 (MCP-1) and tumor necrosis factor-α (TNF-α) assessed by real-time PCR (n = 6 per group). (**C**) Representative images of Immunofluorescence (IF) staining with MPO+ neutrophils. (DAPI: blue;

*Figure 8 continued on next page*

*Figure 8 continued*

MPO: green) Scale bar, 50 μm. (**D**) Representative IF staining images of kidney sections, showing the expression of Cit-H3 (red) and MPO (green). Kidneys were isolated from rIL-19-treated mice after being administered with AAI (n = 6 per group). Scale bar, 50 μm. (**E**) Western blot analysis of Cit-H3. Quantification of the Cit-H3/β-actin ratio (n = 6 per group). (**F–H**) Serum nucleosome, MPO, and dsDNA levels of vehicle and rIL-19-treated mice after being administered with AAI (n = 6 per group). Data are presented as the mean ± SEM of six biological replicates per condition. Each dot represents a sample. Significant differences were determined by one-way ANOVA followed by Tukey's post hoc test (**A, B**, **E–H**). *p<0.05, **p<0.01, ***p<0.001, ns: nonsignificant.

The online version of this article includes the following source data for figure 8:

**Source data 1.** Data represented by each point in *Figure 8A, B, and E–H*.

**Source data 2.** Uncropped western blots for *Figure 8E*.

neutrophil elastase, citrullinated histones, and extracellular DNA (*Papayannopoulos, 2018*). Because PSTPIP2 is closely involved in neutrophil migration, we hypothesized that it participates in the formation and regulation of NETs. In the present study, the levels of NET markers, such as nucleosome, MPO, dsDNA, and Cit-H3, were significantly increased in AAI-exposed mice, and conditional knock-in of *Pstpip2* expression in the kidney reduced NET formation. However, reducing PSTPIP2 expression in mouse kidneys significantly increased NET formation. Indeed, consistent with a previous study, AAN kidneys can also be protected by treatment with the inhibitor of NET formation, GSK484, or DNase I (*Cedervall et al., 2017*), which is consistent with the findings in glomerular disease (*Gupta et al., 2022*).

PSTPIP2 plays an important role in cytokine production. Notably, the expression of IL-19 was the most significantly elevated among the interleukins in AAI-treated mRTECs. Overexpression of PSTPIP2 significantly decreased IL-19 mRNA expression. In the current study, we performed a complementary retrospective association study using serum samples from patients with various kidney diseases. The results indicated that IL-19 was positively correlated with serum Cr level (*Figure 6F*), a classic indicator of kidney injury. IL-19 is a member of the IL-10 family of cytokines, which has indispensable functions in many inflammatory processes. It is also a member of the IL-20 subfamily and binds to the IL-20 receptor to regulate various processes, such as antimicrobial activity, wound healing, and tissue remodeling (*Jennings et al., 2015*). Cellular sources have been characterized as having myeloid and epithelial origins, which correspond with the significant levels found in the RPTEC/TERT1 renal epithelial cell system. IL-19 participates in several animal and human diseases, including psoriasis, inflammatory bowel disease, atherogenesis, endotoxic shock, rheumatoid arthritis, and cancer (*Jennings et al., 2015*). As interleukins play a pivotal role in the formation of NETs, *Yazdani et al., 2017* showed that IL-33 is released from liver sinusoidal endothelial cells to promote NET formation during liver ischemia–reperfusion, exacerbating inflammatory cascades and sterile inflammation *Nie et al., 2019* demonstrated that IL-8-induced NET formation promoted diffuse large B-cell lymphoma progression via TLR9 signaling. IL-19, produced predominantly by osteocytes, stimulates granulopoiesis and neutrophil formation, which in turn stimulates IL-20Rβ/Stat3 signaling in neutrophil progenitors to promote their expansion and neutrophil formation (*Xiao et al., 2021*). To date, the relationship between IL-19 and NET formation has not been reported. In the present study, we found that IL-19 can induce neutrophil chemotaxis and bind to the surface receptor IL-20Rβ to form NETs. Further studies found that PSTPIP2 could affect IL-19 expression by regulating the NF-κB pathway in RTECs. After *Pstpip2* conditional knock-in in mouse kidneys, the expression of IL-19 was inhibited, whereas it was noticeably increased after reducing the expression of PSTPIP2. Therefore, PSTPIP2 affects NET formation precisely because of its regulation of IL-19 release in RTECs.

Finally, it is well accepted that RTEC injury and apoptosis are common histopathological features of AKI (*Havasi and Borkan, 2011*). Although programmed cell death is necessary to maintain health, its dysregulation by excessive or defective apoptosis has been implicated in various diseases (*Poon et al., 2014*). Experimental evidence supports a pathogenic role of apoptosis in AKI. *Wei et al., 2013* reported that inhibition of RTEC apoptosis protected mice from ischemic AKI. We previously found that apoptosis of RTECs contributed to cisplatin nephrotoxicity (*Ma et al., 2017*). We also showed for the first time that histone acetylation-mediated silencing of PSTPIP2 may contribute to cisplatin nephrotoxicity. PSTPIP2 may serve as a potential therapeutic target in the prevention of cisplatin nephrotoxicity. Nevertheless, further studies are needed to determine the mechanism of

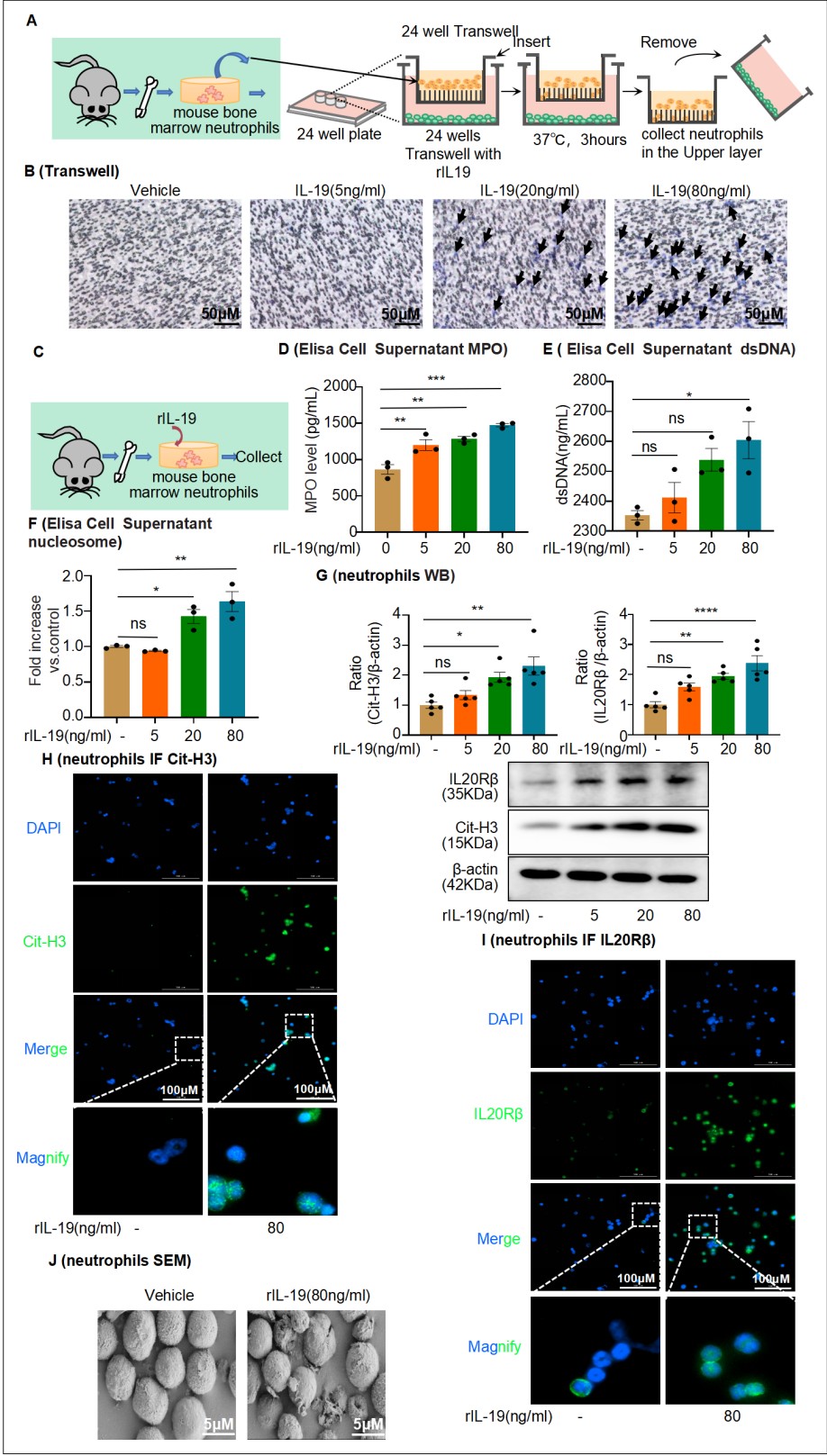

**Figure 9.** Interleukin-19 (IL-19) induces neutrophil extracellular trap (NET) formation in vitro through IL-20Rβ receptor signaling. (**A**) Schematic of Transwell migration assay for neutrophils. (**B**) Neutrophils were isolated from mouse bone marrow and loaded into the upper Transwell chamber. Serum-free RPMI-1640 medium (500 μL) containing 1% penicillin/streptomycin (P/S), with or without different concentrations of IL-19, was added to

*Figure 9 continued on next page*

*Figure 9 continued*

the lower chamber. Neutrophil migration was detected by crystal violet staining. Neutrophils are marked by black arrows (n = 3). Scale bar, 50 µm. (**C**) Mouse bone marrow-derived neutrophils were treated with different concentrations of IL-19 (5 ng/mL, 20 ng/mL, 80 ng/mL) for 4 hr and then collected. (**D, E**) Level of myeloperoxidase (MPO) and dsDNA in the cell culture supernatant assessed after treatment with different concentrations of IL-19 (n = 3). (**F**) Level of nucleosomes in the cell culture supernatant assessed after treatment with different concentrations of IL-19 (n = 3). (**G**) Protein levels of IL-20Rβ and citrullinated histone 3 (Cit-H3) assessed by western blotting after treatment with different concentrations of IL-19 (n = 5). (**H, I**) Immunofluorescence staining of IL-20Rβ and Cit-H3 of neutrophils after treatment with different concentrations of IL-19 (n = 3). Scale bar, 100 µm. (**J**) Representative scanning electron microscopy images of vehicle- and IL-19-stimulated neutrophils (NETs, n = 3). Scale bar, 5 µm. Data are presented as the mean ± SEM of 3–5 biological replicates per condition. Each dot represents a sample. Significant differences were determined by an independent sample *t*-test and one-way ANOVA followed by Tukey's post hoc test (**D–G**). *$p < 0.05$, **$p < 0.01$, ***$p < 0.001$, ****$p < 0.0001$, ns: non-significant.

The online version of this article includes the following source data and figure supplement(s) for figure 9:

**Source data 1.** Data represented by each point in *Figure 9D–G*.

**Source data 2.** Uncropped western blots for *Figure 9G*.

**Figure supplement 1.** Conditional medium (CM) derived from mouse bone marrow-derived neutrophils co-treated with interleukin-19 (IL-19) promotes apoptosis of injured mouse renal tubular epithelial cells (mRTECs).

**Figure supplement 1—source data 1.** Data represented by each point in *Figure 9—figure supplement 1B, C, and E*.

**Figure supplement 1—source data 2.** Uncropped western blots for *Figure 9—figure supplement 1C*.

PSTPIP2-mediated suppression of epithelial cell apoptosis (*Zhu et al., 2020*). NETs are important for immune defense; however, their role in mediating apoptosis remains controversial. In the current study, TUNEL$^+$ cell numbers and caspase-3 activity were significantly increased in the kidneys of mice with AAN, whereas they were significantly decreased in the kidneys of *Pstpip2* conditional knock-in mice. Conversely, the level of apoptosis was markedly increased with a reduction in renal PSTPIP2 expression. We investigated whether NETs also developed during AAI-induced apoptosis and found that the kidneys of neutrophil-depleted mice treated with the Ly6G antibody and NET-depleted mice treated with GSK484 (a PAD4 inhibitor) or DNase I (a nonspecific inhibitor of NETs) showed a significant decrease in the number of TUNEL$^+$ cells. In vitro, we showed that conditioned NET media pretreated with DNase I reduced tubular cell apoptosis. We treated injured tubular epithelial cells by stimulating neutrophil-derived NETs with exogenous IL-19 and found significantly increased levels of apoptosis. Meanwhile, injection of IL-19 protein in mice can aggravate the progression of AAN by increasing renal dysfunction and apoptosis of RTECs.

This study investigated the renoprotective effects of PSTPIP2 in AAI-induced AAN and revealed that NETs participated in both AAI-induced renal injury and apoptosis in RTECs. Moreover, PSTPIP2 reduced IL-19 secretion by inhibiting activation of the NF-κB pathway in RTECs, which in turn reduced the generation of NETs (*Figure 11*).

## Materials and methods
### Reagents and materials

The sources and catalog numbers of the antibodies used were as follows: kidney injury molecule-1 (KIM-1, bs-2713R), PSTPIP2 (bs-19580R), and β-actin (bs-0061R) from Bioss Antibodies (Beijing, China); citrullinated histone H3 (Cit-H3, ab5103), Ly6G (ab238132), interleukin-19 (IL-19, ab154187), and cleaved caspase-3 (ab214430) from Abcam (Cambridge, UK); PSTPIP2 (13450-1-AP), Calbindin-D28k (66394-1-Ig), and E-cadherin (60335-1-Ig) from Proteintech (Wuhan, China); Aquaporin-3 (GB12533-100), Ly-6G (GB11229-100), and cleaved-caspase-3 (GB11532-100) from Servicebio (Wuhan, China); LTL (FL-1321–2) from Vector Laboratories (San Diego, CA); APC-Cy7 rat anti-mouse CD45 (#557659), PE rat anti-mouse CD19 (#557399), BV421 hamster anti-mouse CD3e (#562600), APC rat anti-mouse Ly-6C (#560595), PE rat anti-mouse CD8α (#553032), and 7-AAD (#559925) from BD Biosciences (NJ); myeloperoxidase (MPO, AF3667) from R&D Systems (Minneapolis, MN); IκB-α (T55026), phospho-IκB-α (Ser32/Ser36, TP56280), NF-κB p65 (T55034), and phospho-NF-κB p65 (Ser536, TP56372) from Abmart (Shanghai, China); and IL-20Rβ (#DF3603) and cleaved caspase-3

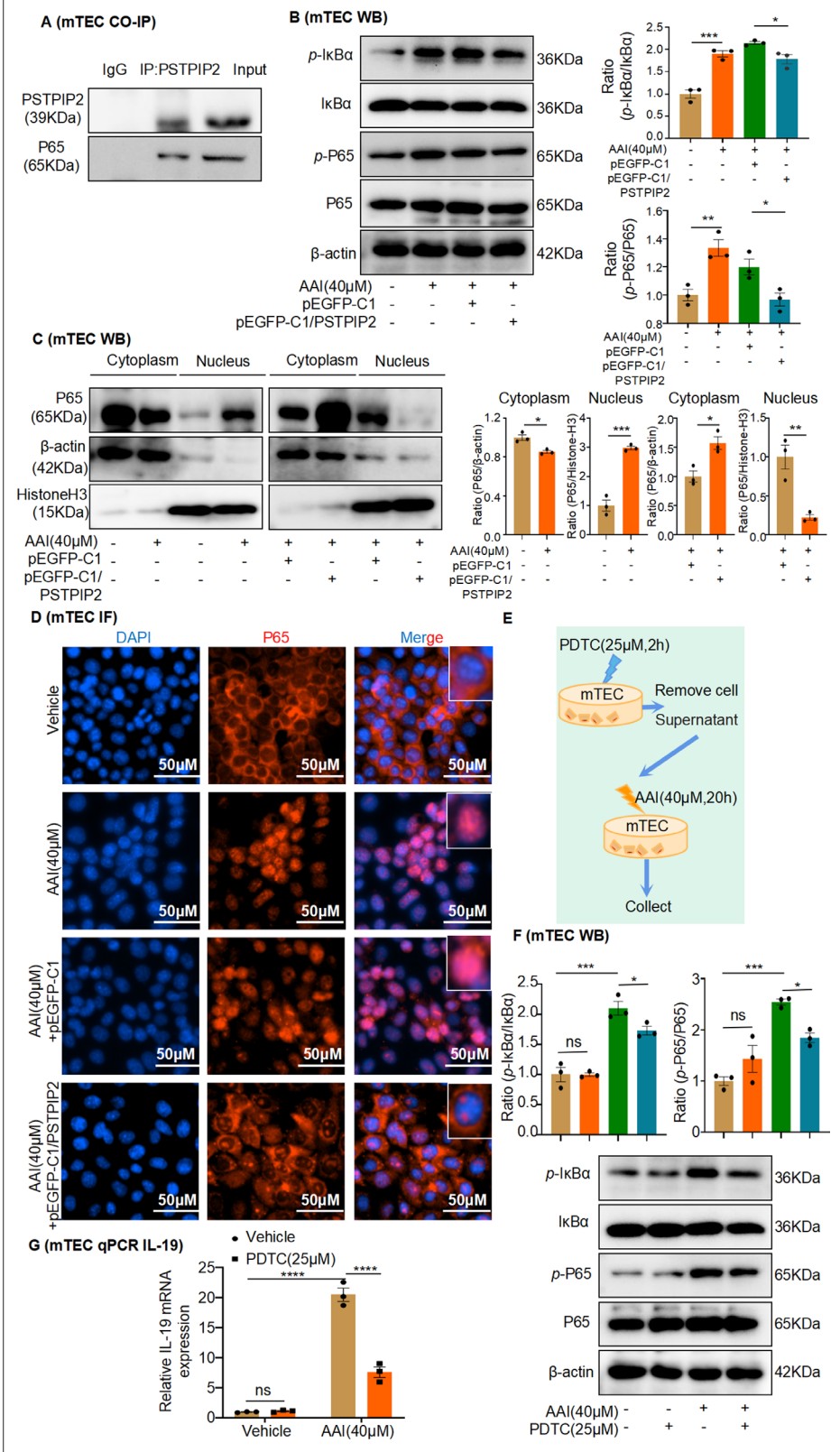

**Figure 10.** PSTPIP2 reduces interleukin-19 (IL-19) transcription by inhibiting NF-κB signaling. (**A**) Co-immunoprecipitation (Co-IP) of PSTPIP2 and NF-κB p65 in mouse renal tubular epithelial cell (mRTEC) using anti-NF-κB p65 and anti-PSTPIP2 antibodies. (**B**) Western blotting results showed that the protein expression of phosphorylated IκBα and p65 was significantly increased after aristolochic acid I (AAI) stimulation. Overexpression

*Figure 10 continued on next page*

*Figure 10 continued*

of PSTPIP2 significantly reduced AAI-induced upregulation of phosphorylated IκBα and p65 proteins (n = 3). (**C**) Expression of NF-κB p65 in the nucleus and cytoplasm of various groups of mRTECs; β-actin and histone H3 were used as cytoplasmic and nuclear protein loading control, respectively (n = 3). (**D**) Immunofluorescence was used to detect the effect of PSTPIP2 overexpression on the nuclear transfer of p65 (n = 3). (**E**) NF-κB signaling was inhibited after a 2 hr pretreatment with the NF-κB inhibitor PDTC (25 μM, n = 3). (**F**) Western blot analysis of phosphorylated IκBα and p65 (n = 3). (**G**) mRNA level of IL-19 in AAI-treated mRTECs exposed to the NF-κB inhibitor PDTC (25 μM, 2 hr) measured by qPCR (n = 3). Data are presented as the mean ± SEM of three biological replicates per condition. Each dot represents a sample. Significant differences were determined by an independent sample *t*-test and one-way ANOVA followed by Tukey's post hoc test (**B, C, F, G**). *p<0.05, **p<0.01, ***p<0.001, ****p<0.0001, ns: nonsignificant.

The online version of this article includes the following source data for figure 10:

**Source data 1.** Data represented by each point in *Figure 10B, C, F, and G*.

**Source data 2.** Uncropped western blots for *Figure 10A, B, C, and F*.

(#AF7022) from Affinity Biosciences (Jiangsu, China). Ultra-LEAF purified anti-mouse Ly6G Antibody (#127649), Ultra-LEAF Purified Rat IgG2a, κ isotype control antibody (#400565), PE anti-mouse/human CD11b antibody (#101208), PE anti-mouse F4/80 (#123109), FITC anti-mouse Ly-6G (#127605), FITC anti-mouse FcεRIα (#134306), APC anti-mouse CD117 (c-Kit) (#105812), and APC anti-mouse CD45 antibody (#103112) were purchased from BioLegend (San Diego, CA). Aristolochic acid I (AAI,

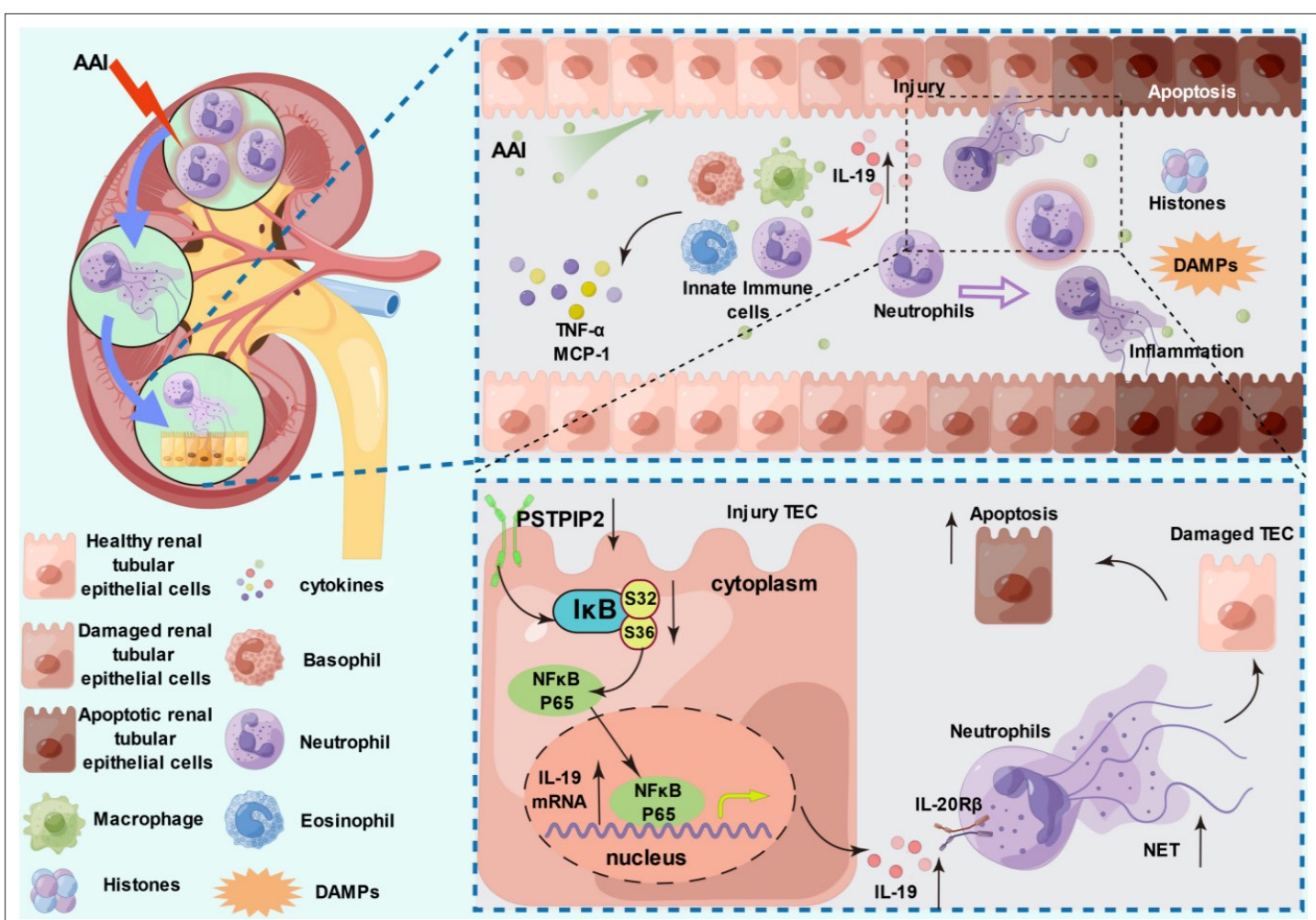

**Figure 11.** PSTPIP2 alleviates aristolochic acid I-induced acute kidney injury and renal tubular epithelial cell apoptosis by suppressing interleukin-19-mediated neutrophil extracellular trap formation.

HY-N0510), GSK484 (HY-100514), and recombinant IL-19 (rIL-19, HY-P7213) were purchased from MedChemExpress (Shanghai, China). DNase I (EN0521) was purchased from Fermentas (Vilnius, Lithuania). Cr and BUN assay kits were purchased from Njjcbio (Nanjing, China). Enzyme-linked immunosorbent assay (ELISA) kits for MPO, MCP-1, tumor necrosis factor (TNF)-α, and IL-19 were purchased from Jymbio (Wuhan, China). An Annexin V-FITC/PI Apoptosis Detection Kit was purchased from BestBio (Shanghai, China). RPMI-1640, Dulbecco's Modified Eagle's Medium/Nutrient Mixture F-12 (DMEM/F-12), and heat-inactivated fetal bovine serum were purchased from Gibco (Carlsbad, CA). Mouse bone marrow neutrophil isolation and caspase-3 activity kits were purchased from Solarbio (Beijing, China). A NETosis Assay Kit was purchased from Cayman Chemical (Ann Arbor, MI). A TUNEL Apoptosis Assay Kit and Nuclear and Cytoplasmic Protein Extraction Kit were purchased from Beyotime Biotechnology (Jiangsu, China). A PicoGreen Assay Kit was purchased from Invitrogen (Carlsbad, CA). A Cell Death kit was purchased from Roche (Mannheim, Germany). An enhanced chemiluminescence (ECL) kit was purchased from Thermo Fisher Scientific (Waltham, MA). A Multi Tissue Dissociation Kit 2 and Anti-Ly6G MicroBeads were purchased from Miltenyi Biotec (Bergisch Gladbach, Germany). D-Luciferin potassium salt and a chemiluminescent luciferase substrate were purchased from Abcam (Cambridge, UK).

## Mouse model of AAN

Six- to eight-week-old male C57BL/6J mice (20–22 g) were obtained from the Experimental Animal Center of the Anhui Medical University. All animal experiments were performed in accordance with the Regulations of the Experimental Animal Administration issued by the State Committee of Science and Technology of China. Efforts were made to minimize the number of animals used and their suffering. Animals were maintained in accordance with the guidance of the Center for Developmental Biology, Anhui Medical University, for the care and use of laboratory animals, and all experiments used protocols approved by the institution's subcommittees on animal care (approval no. LLSC20190682). All mice were randomly assigned to a treatment group and housed in a comfortable environment with a 12 hr/12 hr light/dark cycle for 1 wk before experimentation. Mice were administered AAI (10 mg/kg) via a single intraperitoneal injection, and six mice per group were separately sacrificed at 1, 3, and 5 d post injection. Control mice were injected intraperitoneally with the same dosage of vehicle (10% DMSO + 40% PEG300 + 5% Tween-80 +45% saline).

rIL-19 (carrier-free, 0.2 µg/g mouse; MCE) or vehicle were injected intraperitoneally immediately after AAI (10 mg/kg) administration. All mice were randomly divided into the following four groups (n = 6 per group): vehicle control, model control, IL-19-treated (modeling + intervention with IL-19), and control treated with IL-19 groups. The vehicle group was intraperitoneally injected with vehicle. The AAN model mice were administered AAI (10 mg/kg) via a single intraperitoneal injection and separately sacrificed at 3 d post injection. IL-19-treated mice were injected with IL-19 (0.2 µg/g) immediately after AAI injection. Control treated with IL-19 group mice were only injected with IL-19 (0.2 µg/g). Blood samples and kidney tissues were collected for further analysis.

## Generation of *Pstpip2* conditional knock-in mice and treatment

To create a mouse *Pstpip2* conditional knock-in at the locus of ROSA26 in C57BL/6 mice via CRISPR/Cas-mediated genome engineering. The mouse ROSA26 gene (NCBI Reference Sequence: NR_027008.1) is located on mouse chromosome 6. The mouse *Pstpip2* gene (NCBI Reference Sequence: NM_013831.4) is located on mouse chromosome 18. For the KI model, the 'CAG promoter-loxP-stop-loxP-mouse *Pstpip2* CDS-polyA' cassette was inserted into intron 1 of ROSA26. To engineer the targeting vector, homology arms were generated by PCR using BAC clone from the C57BL/6 library as a template. The gRNA (matching forward strand of gene) sequence was as follows: CTCCAGTCTTTCTAGAAGATGGG. The gRNA to ROSA26 gene, the donor vector containing 'CAG promoter-loxP-stop-loxP-mouse *Pstpip2* CDS-polyA' cassette, and Cas9 mRNA were co-injected into fertilized mouse eggs to generate targeted conditional knock-in offspring. CRISPR/Cas9 was used to form DNA double bond breaks at ROSA26, and then, homologous recombination was used to integrate foreign genes into ROSA26. F0 founder animals were identified by PCR followed by sequence analysis, following which they were bred to wildtype mice to test germline transmission and F1 animal generation. An F1 targeted mouse was bred to a tissue-specific Cdh16-Cre mouse to generate F2. Heterozygous Cre⁺ mice were bred to homozygous mice.

All mice were genotyped by PCR before the experiments. *Pstpip2*$^{Flox/Flox}$ (FF) mouse (Cyagen Biosciences Inc, Guangzhou, China) were hybridized with transgenic mice expressing Cre-recombinase under the cadherin 16 promoter (B6.Cg–Tg(Cdh16-cre)91Igr/J, Jackson Laboratory) to generated tubule-specific *Pstpip2* knock-in mice *Pstpip2*-cKI(Cdh16-Cre$^+$/*Pstpip2*$^{Flox/Flox}$; Cre$^+$/*Pstpip2*$^{Flox/Flox}$). Age-matched mice without Cre (Cdh16-Cre$^-$/*Pstpip2*$^{Flox/Flox}$; Cre$^-$/*Pstpip2*$^{Flox/Flox}$; FF) were used as controls. Flox genotyping produced 372 bp and 607 bp fragments for the mutant and wild type, respectively. Wide type: only a 607 bp band; homozygous (*Pstpip2*$^{Flox/Flox}$): only a 372 bp band; heterozygous (*Pstpip2*$^{Flox/+}$): both bands. A 420 bp band was detected in Cre positive (Cre$^+$), but there is no band in Cre negative (Cre$^-$). Only age-matched male littermates (8–10 weeks old) were used for the experiments; all mice were genotyped by PCR before experimentation. To generate the AAI-induced AAN model, mice were intraperitoneally injected with a single dose of AAI at 10 mg/kg. We sacrificed 6–10 animals per group under anesthesia 3 d after injection. Kidney tissues and blood were collected for further analysis. Blood was collected for BUN and Cr measurements according to the manufacturer's instructions. Kidneys were harvested for paraffin embedding, molecular analysis, and immunostaining.

## AAV9-mediated *Pstpip2* silencing in *Pstpip2*-cKI mice

Luciferase-labeled-specific kidney tissue locations of AAV9-sh*Pstpip2* and the corresponding vector were designed by and obtained from Hanheng (Shanghai, China). All *Pstpip2*-cKI mice were randomly divided into two groups (n = 6 per group): AAV9-NC+AAI and AAV9-sh *Pstpip2*+AAI. Six *Pstpip2*$^{Flox/Flox}$ (FF) mice were randomly selected as the AAI injection group. After 3 wk, a mouse AAN model was established 3 d after AAV9 administration. Mice exposed to AAV9 were anesthetized, and the effect of AAV9-sh*Pstpip2* on kidney tissue localization was confirmed using an IVIS Lumina III Imaging System (Caliper Life Sciences, Hopkinton, MA). AAV9-NC+AAI mice were injected with an empty rAAV9 vector via the tail vein, and AAI was administered at 10 mg/kg via a single intraperitoneal injection. AAV9-sh *Pstpip2*+AAI mice were injected with an AAV9-sh*Pstpip2* vector via the tail vein, and AAI was administered at 10 mg/kg via a single intraperitoneal injection. Mice were sacrificed 72 h after AAI injection, and blood and kidney tissues were collected.

## Neutrophil depletion

Systemic neutrophil depletion was achieved using an IP injection of purified anti-mouse Ly6G (10 µg/g) before and 1 d after the AAI injection. Control mice were treated with the same dosage of rat IgG isotype control, and six mice per group were separately sacrificed at 3 d post injection.

## Neutrophil isolation

Mouse bone marrow-derived neutrophils were isolated and purified by density gradient centrifugation as previously reported (*Li et al., 2018*; *Shi et al., 2020*). Mouse bone marrow-derived neutrophils were extracted as follows: femurs and tibias were aseptically extracted after the animals were sacrificed, and the red marrow cavities were exposed after removing the cartilage at both ends. A 1 mL sterile syringe was used to aspirate a small volume of dilution containing 10% standard fetal bovine serum or serum-containing medium, the marrow cavity was rinsed to obtain marrow, and a single-cell suspension was prepared. Mature neutrophil extraction and purification were performed using the mouse bone marrow neutrophil isolation kit according to the manufacturer's instructions.

To isolate and purify mouse renal tissue-derived primary neutrophils, a single-cell suspension was prepared using a gentleMACS Dissociator and the Multi Tissue Dissociation Kit 2 (Miltenyi Biotec, Cologne, Germany). We separated and purified the neutrophils using Anti-Ly6G MicroBeads (Miltenyi Biotec) magnetic separation with an autoMACS Pro Separator (Miltenyi Biotec) according to the manufacturer's instructions. Neutrophils were extracted and cultured for an additional 1 hr.

## DNase I and GSK484 treatment

DNase I (10 U/100 µL, 0.9% NaCl) or GSK484 (4 mg/kg) was IP injected daily for 3 d prior to AAI treatment, and six mice per group were separately sacrificed at 3 d post-injection.

## NET formation assay

Mouse bone marrow-derived neutrophils were treated with 80 ng/mL IL-19 or DNase I (10 U/mL) for 4 hr. NET formation was assessed using microscopy or ELISA, as described in 'NET formation microscopic analysis' and 'Quantification of NETs'.

## NET extraction and cell culture

Mouse bone marrow-derived neutrophils were treated with 80 ng/mL IL-19 for 4 hr. The primary renal neutrophils of mice subjected to intraperitoneal injections of AAI and DNase I were extracted and cultured for an additional 1 hr. After the cell culture medium was gently aspirated, NETs adhering to the bottom were washed with 2–3 mL cold PBS and centrifuged at $1000 \times g$ for 10 min at 4°C. The cell-free supernatant containing NETs was gently pipetted and stored at −20°C after determining the concentration using PicoGreen.

Mouse renal tubular epithelial cells (mRTECs) were kindly provided by Prof. Huiyao Lan (The Chinese University of Hong Kong) as previously reported (*Jiang et al., 2022a*). Cells were cultured in DMEM/F-12 supplemented with 5% (v/v) heat-inactivated fetal bovine serum at 37°C in a humidified incubator under 5% $CO_2$. The mRTECs were treated with AAI (40 µM) or extracted NETs (500 ng/mL) for 20 hr, which were derived from IL-19-treated mouse bone marrow-derived neutrophils or primary renal neutrophils of mice subjected to IP injection of AAI and DNase I.

## Quantification of NETs

To quantify serum NET levels, we used mouse MPO ELISA kits to detect serum MPO levels. Serum nucleosome quantification was performed using the Cell Death kit. DNA levels in free serum and cell supernatants were quantified using the PicoGreen assay kit according to the manufacturer's instructions.

## Acridine orange–ethidium bromide staining

The morphology of apoptotic-treated cells was detected and distinguished by acridine orange–ethidium bromide fluorescent staining. Staining was performed as previously reported (*Jiang et al., 2022a*).

## Caspase-3 activity assay

The activity of caspase-3 was measured using caspase-3 activity kits. Assays were performed in 96-well microtiter plates, to which 50 µL protein extract, 40 µL reaction buffer, and 10 µL caspase substrate were sequentially added. The protein extracts were incubated at 37°C for 1–2 hr. The absorbance of the samples was measured using a BioTek Cytation 5 Cell Imaging Multimode Reader (BioTek, Winooski, VT) at 405 nm.

## Serum Cr and BUN assays

The serum concentrations of Cr and BUN were determined using the corresponding assay kits according to the manufacturer's instructions.

## Flow cytometry

To assess the level of mRTEC apoptosis, we used an AV-FITC/PI apoptosis detection kit to detect programmed cell death using flow cytometry (Cytoflex, Beckman Coulter, CA). Harvested cells were stained with 400 µL Annexin V binding solution and resuspended. Then, 5 µL annexin V-FITC solution was added, the solution was kept in the dark at 4°C for 15 min, 10 µL propidium iodide was added, and the solution was left for an additional 5 min in the dark at 4°C after gentle mixing. The solution was examined with a flow cytometer and analyzed using CytExpert 2.1 (Beckman Coulter). Neutrophil infiltration (CD11b[+]Ly6G[+]) was analyzed by flow cytometry as previously described (*Huang et al., 2013*).

## TUNEL assay

Renal cell apoptosis was examined by the TUNEL assay using a One-step TUNEL Apoptosis Assay Kit (Beyotime Biotechnology). Briefly, cells were fixed with 4% paraformaldehyde in PBS and exposed to a TUNEL reaction mixture containing TM green-labeled dUTP. The samples were counterstained with

4',6-diamidino-2-phenylindole (DAPI, Servicebio). TUNEL[+] nuclei were identified using fluorescence microscopy (Olympus, Tokyo, Japan).

## Histopathology

Mouse renal tissues were fixed in 4% paraformaldehyde for 24 hr immediately after excision, processed for histological examination according to a conventional method, and stained with H&E (Servicebio). Hematoxylin-stained tissues were visualized and photographed using an automated digital slide scanner (Pannoramic MIDI, 3DHISTECH, Budapest, Hungary).

## ELISA

After blood collection, the serum and erythrocytes were rapidly and carefully separated by centrifugation at 3000 × g for 20 min. The tissues were mashed with the addition of an appropriate amount of PBS. The supernatant was removed by centrifugation at 3000 rpm for 20 min. Serum and tissue monocyte chemoattractant protein-1 (MCP-1) and TNF-α levels were measured using ELISA kits according to the manufacturer's instructions. mRTECs were treated either with AAI (40 µM) or PBS for 20 hr to evaluate the concentration of IL-19 in the supernatant via ELISA. The concentration of IL-19 in the sera of AAN mice was also measured by ELISA using a commercial protocol.

## NET formation microscopic analysis

For microscopic analysis, treated bone marrow-derived neutrophils or renal tissue-derived primary neutrophils were plated on poly L-lysine-coated cover slips. Neutrophils were incubated overnight at 4°C with antibodies against Cit-H3 (1:100) followed by goat anti-rabbit IgG for 1 hr at room temperature (15–25°C). The cells were counterstained with DAPI and visualized using fluorescence microscopy. The levels of neutrophil elastase, which forms part of NETs, were measured using the Cayman Chemical NETosis assay kit according to the manufacturer's specifications. The levels of dsDNA, which forms another part of NETs, were measured using PicoGreen according to the manufacturer's specifications.

## Immunohistochemistry

Kidney tissues were fixed in 4% paraformaldehyde for 24 hr immediately after excision, embedded in paraffin, and cut to 5-µm-thick sections. The sections were exposed to 3% hydrogen peroxide for 10 min to suppress endogenous peroxidase activity and then blocked with 3% bovine serum albumin (BSA; Servicebio) for 30 min. The tissue sections were incubated with anti-Ly6G (1:2000), KIM-1 (1:200), and cleaved caspase-3 (1:100) antibodies overnight at 4°C. The sections were then washed and incubated with anti-rabbit IgG secondary antibodies for 30 min at room temperature (15–25°C). The sections were counterstained with hematoxylin and developed using a diaminobenzidine (DAB) kit. The stained sections were photographed and visualized using the Pannoramic MIDI (3DHISTECH).

## Immunofluorescence staining

Mouse renal tissue sections were blocked with 5% BSA at 37°C for 30 min to avoid non-specific staining. Sections were incubated with anti-Cit-H3 (1:100), anti-MPO (1:100), anti-IL-19 (1:100), anti-E-cadherin (1:100), anti-LTL(1:80), anti-calbindin D28k (1:150), anti- aquaporin-3 (1:800), or anti-PSTPIP2 (1:400) antibodies overnight at 4°C. Then, the sections were incubated with a secondary antibody (1:100) in the dark at 37°C for 1 hr. After the nuclei were stained with DAPI, the stained sections were examined using inverted fluorescence microscopy (Olympus).

After discarding the cell culture medium, mRTECs were washed twice with PBS, blocked with 5% BSA for 30 min after fixation and permeabilization, and incubated with anti-KIM-1 (1:100), PSTPIP2 (1:100), IL-19 (1:100), and NF-κB p65 (1:50) antibodies at 4°C overnight. Neutrophils were incubated overnight at 4°C with antibodies against Cit-H3 (1:100) and IL-20Rβ (1:200) antibodies. After washing with PBS, cells were incubated with the corresponding fluorescent secondary antibodies for 1 hr at room temperature (15–25°C). After the nuclei were stained with DAPI (Servicebio), the sections were blocked with an anti-fluorescent quencher-containing blocking solution and observed under a fluorescence microscope (Olympus).

## Western blot analysis

Cells were lysed with radio immunoprecipitation assay (RIPA) lysis buffer to extract proteins from mouse kidney tissues (20 or 40 mg), mRTECs, and neutrophils. The Nuclear and Cytoplasmic Protein

Extraction Kit (Beyotime Biotechnology) was used to extract nuclear and cytoplasmic proteins from $6 \times 10^5$ mRTECs, according to the manufacturer's instructions. Whole extracts were separated using 8–12% sodium dodecyl sulfate polyacrylamide gel electrophoresis (SDS-PAGE), transferred to a polyvinylidene fluoride (PVDF) membrane (Millipore, Billerica, MA), blocked with 5% non-fat dry milk (NFDM) for 2 hr at room temperature (15–25 °C), and washed thrice with Tris-buffered saline solution. PVDF membranes were incubated with primary antibodies against KIM-1, cleaved caspase-3, Cit-H3, PSTPIP2, IL-20Rβ, NF-κB p65, phospho-NF-κB p65 (Ser536), IκB-α, phospho-IκB-α (Ser32/Ser36), histone H3, and β-actin. The membranes were washed with TBS-Tween 20 and incubated with secondary antibodies. After extensive washing in TBS-Tween 20, protein bands were visualized with the ECL kit (Epizyme, Shanghai, China). The signal intensity of each protein band was quantified using ImageJ software (National Institutes of Health, Bethesda, MD).

## Real-time reverse transcription-PCR

Total RNA was extracted from kidney tissues using TRIzol reagent (Invitrogen). First-strand cDNA was synthesized using a ThermoScript RT-PCR synthesis kit (AG, Hunan, China), according to the manufacturer's instructions. Real-time quantitative PCR analysis of mRNA was performed using ThermoScript RT-qPCR kits in an ABI Prizm step-one plus real-time PCR system (Applied Biosystems, Foster City, CA). The products were used as templates for amplification using SYBR Green PCR amplification reagent and gene-specific primers. The relative expression levels were calculated using the standard $2^{-\Delta\Delta Ct}$ method. The forward and reverse primers used for PCR are listed in *Table 1*.

## Clinical specimens

The clinical features of patients with various kidney diseases are listed in *Table 2*. This study was approved by the Biomedical Ethics Committee of the Anhui Medical University (ethical clearance no.: 83230373). The experiments were conducted with the understanding and written consent of each patient.

## Transwell migration assay of neutrophil levels

Isolated bone marrow-derived neutrophils were resuspended in serum-free RPMI-1640 medium containing 1% penicillin/streptomycin (P/S) at $10^6$/mL. Then, 200 µL of cells were seeded in the upper well of a Transwell chamber with microporous filters (3 µm pores; Corning, Corning, NY), and 500 µL of 1% P/S serum-free RPMI-1640 medium in the absence or presence of IL-19 (80 ng/mL) was added to the bottom chamber. After incubation for 3 hr, cells adherent to the membrane in a 3 µm pore Transwell system were fixed with 4% paraformaldehyde for 10 min, stained with 1% crystal violet for 30 min, and observed under a light microscope (Olympus).

## Transfection with PSTPIP2 plasmid

mRTECs ($2–4 \times 10^5$ cells) were seeded into a 6-well plate and cultured in DMEM/F-12 medium (containing 5% FBS) until cell attachment and grown to a density of ~50%. Cells were transfected with 2000 ng/mL pEGFP-C1or pEGFP-C1/PSTPIP2 overexpression plasmid mixed with a Lipo3000 transfection reagent (Hanheng). After 6 hr, the Opti-MEM was replaced with DMEM/F-12 containing 5% FBS, and the cells were treated with AAI (40 µM) for 20 hr. Transfection efficiency was determined using western blotting and real-time PCR.

## Co-immunoprecipitation

Cells were collected after washing with pre-chilled PBS solution and an appropriate amount of pre-chilled lysis buffer was added. After ultrasonic disruption and centrifugation, the cell supernatants were collected and stored. PSTPIP2 and NF-κB p65 antibodies (1–5 µg) were added to the samples, along with a nonspecifically immunized cognate antibody as a control. After overnight incubation at 4°C, 5 µL of protein A-agarose beads and 5 µL protein G-agarose beads were added to the samples and mixed gently overnight at 4°C. Immunocomplex pellets were obtained after centrifugation, resuspended in SDS buffer, boiled, stored, and resolved in 8–12% SDS-polyacrylamide gels for western blot analysis with PSTPIP2 and NF-κB p65 antibodies.

**Table 1.** Primer sequences for the quantitative real-time PCR analysis of mouse renal tubular epithelial cells (mRTECs) and mouse tissues.

| Terms (mouse) | Forward primer (5'–3') | Reverse primer (5'–3') |
| --- | --- | --- |
| IL1α | TCTATGATGCAAGCTATGGCTCA | CGGCTCTCCTTGAAGGTGA |
| IL1β | GAAATGCCACCTTTTGACAGTG | TGGATGCTCTCATCAGGACAG |
| IL3 | GGGATACCCACCGTTTAACCA | AGGTTTACTCTCCGAAAGCTCTT |
| IL4 | GGTCTCAACCCCCAGCTAGT | GCCGATGATCTCTCTCAAGTGAT |
| IL5 | GCAATGAGACGATGAGGCTTC | GCCCCTGAAAGATTTCTCCAATG |
| IL6 | CTGCAAGAGACTTCCATCCAG | AGTGGTATAGACAGGTCTGTTGG |
| IL7 | TTCCTCCACTGATCCTTGTTCT | AGCAGCTTCCTTTGTATCATCAC |
| IL9 | ATGTTGGTGACATACATCCTTGC | TGACGGTGGATCATCCTTCAG |
| IL10 | CTTACTGACTGGCATGAGGATCA | GCAGCTCTAGGAGCATGTGG |
| IL11 | GCGCTGTTCTCCTAACCCG | GAGTCCAGACTGTGATCTCCG |
| IL12α | CAATCACGCTACCTCCTCTTTT | CAGCAGTGCAGGAATAATGTTTC |
| IL13 | TGAGCAACATCACACAAGACC | GGCCTTGCGGTTACAGAGG |
| IL14 | TCCTGAGTACATACTGTGTGGAC | GCTGCATAGGTTCGGGACTTC |
| IL15 | CATCCATCTCGTGCTACTTGTG | GCCTCTGTTTTAGGGAGACCT |
| IL16 | AAGAGCCGGAAATCCACGAAA | GTGCGAGGTCTGGGATATTGC |
| IL17A | TCAGCGTGTCCAAACACTGAG | CGCCAAGGGAGTTAAAGACTT |
| IL17F | TGCTACTGTTGATGTTGGGAC | CAGAAATGCCCTGGTTTTGGT |
| IL18 | GTGAACCCCAGACCAGACTG | CCTGGAACACGTTTCTGAAAGA |
| IL19 | CTCCTGGGCATGACGTTGATT | GCATGGCTCTCTTGATCTCGT |
| IL20 | GTCTTGCCTTTGGACTGTTCT | AGGTTTGCAGTAATCACACAGC |
| IL21 | GGACCCTTGTCTGTCTGGTAG | TGTGGAGCTGATAGAAGTTCAGG |
| IL22 | ATGAGTTTTTCCCTTATGGGGAC | GCTGGAAGTTGGACACCTCAA |
| IL23 | CAGCAGCTCTCTCGGAATCTC | TGGATACGGGGCACATTATTTTT |
| IL24 | GAGCCTGCCCAACTTTTTGTG | TGTGTTGAAGAAAGGGCCAGT |
| IL25 | ACAGGGACTTGAATCGGGTC | TGGTAAAGTGGGACGGAGTTG |
| IL27 | CTGTTGCTGCTACCCTTGCTT | CTCCTGGCAATCGAGATTCAG |
| IL28B | GTTCAAGTCTCTGTCCCCAAAA | GTGGGAACTGCACCTCATGT |
| IL31 | TCAGCAGACGAATCAATACAGC | TCGCTCAACACTTTGACTTTCT |
| IL33 | ATTTCCCCGGCAAAGTTCAG | AACGGAGTCTCATGCAGTAGA |
| IL34 | TTGCTGTAAACAAAGCCCCAT | CCGAGACAAAGGGTACACATTT |
| IL40 | ACTGGAAGTTTATCCCCAAAGC | CGGAGTCATGCACAACCTTTTT |
| Kim-1 | TAAACCAGAGATTCCCACAC | GATCTTGTTGAAATAGTCGTG |
| MCP-1 | GCTTGAGGTGGTTGTGGAAAA | CTCACCTGCTGCTACTCATTC |
| β-Actin | GATTACTGCTCTGGCTCCTAGC | GACTCATCGTACTCCTGCTTG |
| TNF-α | CCCTCACACTCAGATCATCTTCT | GCTACGACGTGGGCTACAG |

**Table 2.** Clinical features of the patients with kidney diseases.

| No. | Pathological diagnosis | Sex | Age (years) | Serum creatinine (μmol/L) | Serum IL-19 (ng/mL) |
|---|---|---|---|---|---|
| 1 | Kidney transplant | Female | 19 | 198.4278 | 274.4815 |
| 2 | Kidney transplant | Female | 33 | 144.0580 | 297.7699 |
| 3 | Hydronephrosis with ureteral stones | Male | 48 | 216.9878 | 125.0563 |
| 4 | Hydronephrosis with nephrolithiasis | Male | 47 | 158.1964 | 199.1192 |
| 5 | Renal allograft dysfunction | Male | 27 | 450.1882 | 308.2646 |
| 6 | Allograft dysfunction, renal allograft | Male | 36 | 444.1835 | 161.4381 |
| 7 | Renal anemia, 5 CKD stage | Male | 58 | 1225.2303 | 340.7484 |
| 8 | CKD stage 3, type 2 diabetes | Female | 76 | 129.5376 | 199.1192 |
| 9 | Nephrotic syndrome with minimal change nephropathy | Female | 21 | 160.7620 | 231.6030 |
| 10 | Renal end-stage disease | Male | 35 | 1250.3954 | 391.4230 |
| 11 | Nephrotic syndrome | Male | 56 | 124.0242 | 240.0487 |
| 12 | Polycystic kidney disease | Female | 57 | 210.0129 | 147.1452 |
| 13 | Renal end-stage disease | Female | 38 | 1229.0289 | 468.0847 |
| 14 | Chronic renal failure | Male | 57 | 807.2540 | 218.6095 |
| 15 | Hydronephrosis with nephroureterolithiasis, type I diabetes | Female | 53 | 83.8521 | 226.4056 |
| 16 | Diabetes | Male | 66 | 82.1029 | 229.0043 |
| 17 | Nephrolithiasis with hydronephrosis | Male | 34 | 139.9357 | 208.2147 |
| 18 | CKD stage 4 | Male | 55 | 134.3601 | 201.7179 |
| 19 | Membranous nephropathy | Male | 40 | 145.3473 | 253.6919 |
| 20 | Nephrotic syndrome | Male | 46 | 76.5273 | 262.7874 |
| 21 | Hydronephrosis | Female | 52 | 92.5981 | 135.4511 |
| 22 | Renal insufficiency | Male | 82 | 754.3408 | 225.1062 |
| 23 | SLE, lupus nephritis | Male | 32 | 456.8682 | 391.4230 |
| 24 | Hydronephrosis with nephrolithiasis | Female | 68 | 128.7846 | 266.6854 |
| 25 | Diabetes | Male | 43 | 107.3569 | 283.5770 |
| 26 | Membranous nephropathy | Male | 53 | 174.3729 | 442.0977 |
| 27 | Nephrotic syndrome | Male | 46 | 186.9453 | 188.7244 |

## Statistical analysis

Data are expressed as the mean ± standard error of the mean (SEM) from at least three independent experiments. Differences between two groups were compared using a two-tailed Student's $t$-test. Differences between multiple groups were compared using a one-way ANOVA followed by Tukey's post hoc test. Differences were considered significant at $p < 0.05$. All analyses were performed using GraphPad Prism (version 8.0; GraphPad Software, San Diego, CA).

## Study approval

The in vivo experiments were approved by the Anhui Medical University's Institutional Animal Care and Use Committee (approval No. LLSC20190682) and performed according to institutional animal care guidelines. The experiments were conducted in Association for Assessment and Accreditation of Laboratory Animal Care-accredited facilities.

## Acknowledgements

The authors thank the Center for Scientific Research of Anhui Medical University for valuable help in our experiment. *Figure 11* were created by Figdraw.

## Additional information

### Funding

| Funder | Grant reference number | Author |
|---|---|---|
| Department of Science and Technology of Anhui Province | Natural Science Foundation of Anhui Province(2008085MH273) | Taotao Ma |
| Department of Science and Technology of Anhui Province | Anhui Fund for Distinguished Young Scholars(2022AH020050) | Taotao Ma |
| Anhui Medical University | Research Fund of Anhui Institute of translational medicine(2022zhyx-B07) | Taotao Ma |
| Anhui Medical University | Research Fund of Anhui Institute of translational medicine(2021zhyx-B06) | Cheng Huang |
| Anhui Medical University | Scientific Research Promotion Fund of Anhui Medical University (2022xkjT010) | Cheng Huang |
| Anhui Medical University | Scientific Research Platform Improvement Project of Anhui Medical University (2023xkjT049) | Jun Li |

The funders had no role in study design, data collection and interpretation, or the decision to submit the work for publication.

### Author contributions

Changlin Du, Conceptualization, Data curation, Software, Formal analysis, Methodology, Writing - original draft; Chuanting Xu, Resources, Data curation, Validation, Visualization, Methodology; Pengcheng Jia, Resources, Software, Supervision; Na Cai, Data curation, Investigation, Visualization, Methodology; Zhenming Zhang, Validation, Visualization, Methodology; Wenna Meng, Lu Chen, Supervision, Investigation, Methodology; Zhongnan Zhou, Investigation, Methodology; Qi Wang, Resources, Software, Supervision, Methodology; Rui Feng, Supervision, Investigation, Methodology, Project administration; Jun Li, Conceptualization, Resources, Supervision, Funding acquisition, Project administration; Xiaoming Meng, Resources, Investigation, Visualization, Methodology; Cheng Huang, Conceptualization, Resources, Data curation, Supervision, Funding acquisition, Investigation, Project administration, Writing - review and editing; Taotao Ma, Conceptualization, Resources, Supervision, Funding acquisition, Project administration, Writing - review and editing

### Author ORCIDs

Taotao Ma (iD) http://orcid.org/0000-0003-2208-2505

### Ethics

This study was approved by the Biomedical Ethics Committee of the Anhui Medical University (ethical clearance no.: 83230373). The experiments were conducted with the understanding and written consent of each patient.

All animal experiments were performed in accordance with the Regulations of the Experimental Animal Administration issued by the State Committee of Science and Technology of China. Efforts were made to minimize the number of animals used and their suffering. Animals were maintained in accordance with the guidance of the Center for Developmental Biology, Anhui Medical University, for

the care and use of laboratory animals, and all experiments used protocols approved by the institutions' subcommittees on animal care. (approval No. LLSC20190682).

### Decision letter and Author response
Decision letter https://doi.org/10.7554/eLife.89740.sa1
Author response https://doi.org/10.7554/eLife.89740.sa2

---

## Additional files

### Supplementary files
• MDAR checklist

### Data availability
All data generated or analyzed during this study are included in the manuscript and supporting files.

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
