## [Editor Report]

This study presents valuable new insights to support Netosis plays an important role in the development of aristolochic acid nephropathy (AAN). A series of compelling experiments using in vivo and in vitro model supported that AAN induced NET formation via IL-19-IL20-β receptor can induce inflammation and cell death. This new knowledge of the interaction between kidney cells and neutrophils could have clinical implications in the treatment of AAN.

---

## [Decision Letter]

**Decision letter after peer review:**

Thank you for submitting your article "PSTPIP2 ameliorates aristolochic acid nephropathy by suppressing interleukin-19-mediated neutrophil extracellular trap formation" for consideration by *eLife*. Your article has been reviewed by 2 peer reviewers, and the evaluation has been overseen by a Reviewing Editor and Satyajit Rath as the Senior Editor. The reviewers have opted to remain anonymous.

Essential revisions:

1) Important to depict the cell-specific location of PSTPIP2 expression in various kidney cells.

2) More mechanistic studies to support findings in the mouse model should provide more clarity on the key pathways for potential future translational studies.

*Reviewer #1 (Recommendations for the authors):*

In this study, the authors identified the functional role of PSTPIP2 in aristolochic acid nephropathy. Mechanistically, they demonstrated that PSTPIP2 affected NF-κB signaling to inhibit IL-19 production, which could induce Netosis, finally limiting inflammation and kidney injury. Some comments have to be addressed to support the conclusion.

1. This study didn't investigate the function of IL-19 in aristolochic acid nephropathy in vivo. Whether injection of IL-19 protein accelerates aristolochic acid nephropathy? In addition, this study mainly explores the role of PSTPIP2 in aristolochic acid nephropathy. The authors should introduce PSTPIP2 in the INTRODUCTION part.

2. In Figure 2A, the genetic approach for generating PSTPIP2 KI mice seems to be knockout mice. the authors should carefully prepare this figure. Importantly, as described in the method, the "CAG-promoter-loxP-stop-loxP PSTPIP2" cassette was inserted into the fertilized mouse eggs, thus PSTPIP2 should not be detected in tissues. But PSTPIP2 was detected as shown in Figure 2C.

3. Where does PSTPIP2 express in cells? in Cytoplasm or in Nucleus? Because some staining of PSTPIP2 is not consistent. How about IL-19 staining? in Cytoplasm or in Nucleus?

4. The analysis for the population of neutrophils by flow is not convincing. The pattern for CD11b+GR1+ cells is not correct. Please double-check this important data. In addition, for the quantification of Ly6G, the authors should use Ly6G+ cells, not positive area.

5. It's difficult to believe that IL-19 promotes neutrophil infiltration as shown in Figure 7B. The authors should provide more solid data.

6. Authors should check and review extensively for improvements to the use of English.

*Reviewer #2 (Recommendations for the authors):*

This study by Du et. al addressed the role and regulation of proline-serine-threonine phosphatase interacting protein 2 (PSTPIP2) and neutrophil extracellular traps (NETs) in Aristolochic acid Nephropthathy (AAN) and immune defense. PSTPIP2 expression is downregulated in AAN. Conditional knock-in of PSTPIP2 in mouse kidneys inhibited cell apoptosis, reduced neutrophil infiltration, suppressed the production of inflammatory factors and NETs, and ameliorated renal dysfunction. Reducing the expression of PSTPIP2 to normal levels in knock-in mouse using shRNA promoted kidney injury. Using in vivo model, the role of PSTPIP2 in AAN injury and renal function, apoptosis, neutrophil infiltration and NET formation is established. Using in vitro models, a PSTPIP2/NFkB-mediated NET formation via IL-19-IL20-β Receptor pathway is shown to induce inflammation and apoptosis in AAN. The studies are well presented.

1. The studies could be strengthened by replicating the in vitro mechanistic pathway in the mouse model.

2. The abstract should include a concluding sentence.

---

## [Author Response]

Essential revisions:1) Important to depict the cell-specific location of PSTPIP2 expression in various kidney cells.

Thank you for your comment.

To explore the location of PSTPIP2 in the setting of Aristolochic acid I(AAI)-induced AAN or PSTPIP2 expressed in which segments of the nephron, immunofluorescence of lotus tetragonolobus lectin (LTL) (a proximal tubule marker), calbindin D28k (a distal tubule marker), aquaporin-3 (a collecting duct marker), and PSTPIP2 was perfomed. As a result, AAI markedly downregulated PSTPIP2 expression in distal tubule epithelial cells and collecting ducts.

2) More mechanistic studies to support findings in the mouse model should provide more clarity on the key pathways for potential future translational studies.

Thank you for your comment.

To confirm whether IL-19 can accelerate aristolochic acid nephropathy and to identify the functional role of IL-19 in NET formation in mice intraperitoneally injected with AAI, recombinant mouse IL-19 (rIL-19) was administered to mice immediately after treatment with AAI. Histological analysis with H&E staining revealed that intraperitoneal injection of rIL-19 at a dose of 4 µg caused no damage in vehicle-treated mice but considerably accelerated renal damage when combined with AAI treatment (Figure 7A). Similarly, intraperitoneal injection of IL-19 significantly increased the levels of Cr and BUN in the serum of AAI-treated mice (Figure 7B and C). In addition, western blotting and PCR and IHC analysis showed that KIM-1 expression was significantly upregulated in IL-19-treated AAN mice compared with vehicle-treated AAN mice (Figure 7D-F). Furthermore, we found that rIL-19 significantly increased TUNEL^+^ cell numbers (Figure 7I) and significantly upregulated the activity of caspase-3 and level of cleaved caspase-3 following intraperitoneal injection of IL-19 in AAI-treated mice (Figure 7G and H). These results suggest that injection of IL-19 protein accelerates aristolochic acid nephropathy. Intraperitoneal injection with AAI resulted in increased numbers of MPO^+^ neutrophils (Figure 8C) in the injured kidney, as well as increased serum levels of TNF-α and MCP-1 in rIL-19-treated mice compared to vehicle-treated mice. (Figure 8A and B). Similarly, NET formation, as measured by serum levels of MPO, dsDNA, and nucleosomes, as well as by tissue levels of Cit-H3 in the injured kidney, significantly increased in rIL-19-treated AAN mice compared to vehicle-treated AAN mice (Figure 8D-H). These results also indicated that IL-19 could further drive the inflammatory response and NET formation in aristolochic acid nephropathy.

Reviewer #1 (Recommendations for the authors):In this study, the authors identified the functional role of PSTPIP2 in aristolochic acid nephropathy. Mechanistically, they demonstrated that PSTPIP2 affected NF-κB signaling to inhibit IL-19 production, which could induce Netosis, finally limiting inflammation and kidney injury. Some comments have to be addressed to support the conclusion.1. This study didn't investigate the function of IL-19 in aristolochic acid nephropathy in vivo. Whether injection of IL-19 protein accelerates aristolochic acid nephropathy? In addition, this study mainly explores the role of PSTPIP2 in aristolochic acid nephropathy. The authors should introduce PSTPIP2 in the INTRODUCTION part.

Thank you for your comment.

We appreciate it very much for this good suggestion, and we have done it according to your ideas. We investigated the effect of IL-19 on Aristolochic Acid Nephropathy in mice by intraperitoneal injection of exogenous recombinant IL-19 (200ug/g mice). rIL-19 (carrier-free, 0.2 µg/g mouse; MCE) or vehicle were injected intraperitoneally immediately after AAI (10 mg/kg) administration. All mice were randomly divided into the following four groups (n = 6 per group): vehicle control, model control, IL-19-treated (modeling + intervention with IL-19), and control treated with IL-19 groups. The vehicle group was intraperitoneally injected with vehicle. The Aristolochic acid nephropathy model mice were administered AAI (10 mg/kg) via a single intraperitoneal injection and separately sacrificed at 3-d post injection. IL-19-treated mice were injected with IL-19 (0.2 µg/g) immediately after AAI injection. Control treated with IL-19 group mice were only injected with IL-19 (0.2 µg/g). Blood samples and kidney tissues were collected for further analysis. rIL-19 at a dose of 4ug injected intraperitoneally caused no damage in vehicle-treated mice but drastically accelerated renal damage when treated with AAI, based on a histological analysis with H&E staining (Figure 7A). Similarly, intraperitoneal injection of IL-19 significantly increased the levels of Cr and BUN in the serum of AAI-treated mice(Figure 7B and C). In addition, western blot, PCR and IHC analysis showed that KIM-1 expression was significantly upregulated in IL-19-treated AAN mice compare with vehicle-treated AAN mice (Figure 7D-F). Furthermore, we found that rIL-19 significantly increased TUNEL^+^ cell numbers (Figure 7I) and significantly upregulated the activity of caspase-3 and level of cleaved caspase-3 following intraperitoneal injection of IL-19 in AAI-treated mice (Figure 7G and H). These results suggest that injection of IL-19 protein accelerates aristolochic acid nephropathy.

At the same time, we added the description of PSTPIP2 in the paragraph of Introduction and line 96-109.

“Proline-serine-threonine phosphatase-interacting protein 2 (PSTPIP2), belonging to the Fes/CIP4 homology-bin/amphiphysin/rvs (F-BAR) family, or the pombe cdc15 homology family proteins (18), is situated on chromosome 18 in both mice and humans (19). PSTPIP2 is implicated in immunological and autoinflammatory diseases (20) and is expressed not only in various tissues and organs (e.g., heart, liver, lungs) but also in monocytes, mast cells, lymphocytes, and granulocytes (21). Studies have demonstrated that PSTPIP2 significantly contributes to inflammatory disorders, tumors, and various diseases by modulating cell proliferation, apoptosis, and the secretion of inflammatory factors (22, 23). Wang et al. showed that PSTPIP2 is associated with sepsis and can regulate the expression of inflammatory factors by modulating the NF-κB pathway (24). Pavliuchenko et al. reported that molecular interactions involving the adaptor protein PSTPIP2 control neutrophil-mediated responses, leading to autoinflammation (25). Furthermore, in a previous study, we identified a pivotal role of pstpip2 in cisplatin-induced acute kidney injury (21).”

2. In Figure 2A, the genetic approach for generating PSTPIP2 KI mice seems to be knockout mice. the authors should carefully prepare this figure. Importantly, as described in the method, the "CAG-promoter-loxP-stop-loxP PSTPIP2" cassette was inserted into the fertilized mouse eggs, thus PSTPIP2 should not be detected in tissues. But PSTPIP2 was detected as shown in Figure 2C.

Thank you for your comment.

We carefully checked Figure2A and modified it as follows. At the same time, we supplemented the construction of kidney conditional knock in mice of *Pstpip2* in the Materials and methods part of the article. At the same time, for Figure 2C in the article, we detected the mice that have been genotyped by PCR. The offspring mice generated by the hybridization of *Pstpip2*^flox/flox^ (FF) mice and Cdh16-Cre^+^/*Pstpip2*^flox/flox^ (*Pstpip2*-cKI) mice, and these were not F0 and F1 mice.

“To create a mouse *Pstpip2* conditional knock-in at the locus of ROSA26 in C57BL/6 mice via CRISPR/Cas-mediated genome engineering. The mouse ROSA26 gene (NCBI Reference Sequence: NR_027008.1) is located on mouse chromosome 6. The mouse *Pstpip2* gene (NCBI Reference Sequence: NM_013831.4) is located on mouse chromosome 18. For the KI model, the “CAG promoter-loxP-stop-loxP-mouse *Pstpip2* CDS-polyA” cassette was inserted into intron 1 of ROSA26. To engineer the targeting vector, homology arms were generated by PCR using BAC clone from the C57BL/6 library as a template. The gRNA (matching forward strand of gene) sequence was as follows: CTCCAGTCTTTCTAGAAGATGGG. The gRNA to ROSA26 gene, the donor vector containing “CAG promoter-loxP-stop-loxP-mouse *Pstpip2* CDS-polyA” cassette, and Cas9 mRNA were co-injected into fertilized mouse eggs to generate targeted conditional knock-in offspring. CRISPR/Cas9 was used to form DNA double bond breaks at ROSA26, and then, homologous recombination was used to integrate foreign genes into ROSA26. F0 founder animals were identified by PCR followed by sequence analysis, following which they were bred to wildtype mice to test germline transmission and F1 animal generation. An F1 targeted mouse was bred to a tissue-specific Cdh16-Cre mouse to generate F2. Heterozygous Cre^+^ mice were bred to homozygous mice.

All mice were genotyped by PCR before the experiments. *Pstpip2*^Flox/Flox^ (FF) mouse (Cyagen Biosciences Inc, Guangzhou, China) were hybridized with transgenic mice expressing Cre-recombinase under the cadherin 16 promoter (B6.Cg–Tg(Cdh16-cre)91Igr/J, Jackson Laboratory) to generated tubule-specific *Pstpip2* knockin mice *Pstpip2*-cKI(Cdh16-Cre^+^/*Pstpip2*^Flox/Flox^; Cre^+^/*Pstpip2*^Flox/Flox^). Age-matched mice without Cre (Cdh16-Cre^-^/*Pstpip2*^Flox/Flox^; Cre^-^/*Pstpip2*^Flox/Flox^; FF) were used as controls. Flox genotyping produced 372 bp and 607 bp fragments for the mutant and wild type respectively. Wide type: only a 607 bp band; homozygous (*Pstpip2*^Flox/Flox^): only a 372 bp band; heterozygous (*Pstpip2*^Flox/+^): both bands. A 420 bp band was detected in Cre positive (Cre^+^), but there is no band in Cre negative (Cre^−^). to homozygous mice.” As shown in Author response image 1:

**Author response image 1. sa2fig1:** 

3. Where does PSTPIP2 express in cells? in Cytoplasm or in Nucleus? Because some staining of PSTPIP2 is not consistent. How about IL-19 staining? in Cytoplasm or in Nucleus?

Thank you for your comment.

According to relevant literature reports and experimental demonstration, PSTPIP2 is mainly expressed in the cell membrane (Zhu et al., 2020), while IL-19 is a secreted protein, which is mainly synthesized in the cell and secreted outside the cell (Sabat et al.,2007). We carefully checked and corrected the immunofluorescence staining pictures of PSTPIP2 and IL-19 in the experimental results (Figure1D, Figure 6—figure supplement 1B-C), and observed the expression of PSTPIP2 and IL-19 in the cells by using a laser confocal microscope (Figure1H, 6E, Figure 6—figure supplement 1F).

Zhu H, Jiang W, Zhao H, et al. PSTPIP2 inhibits cisplatin-induced acute kidney injury by suppressing apoptosis of renal tubular epithelial cells. Cell Death Dis. 2020;11(12):1057.

Sabat R, Wallace E, Endesfelder S, Wolk K. IL-19 and IL-20: two novel cytokines with importance in inflammatory diseases. Expert Opin Ther Targets. 2007;11(5):601-612.

4. The analysis for the population of neutrophils by flow is not convincing. The pattern for CD11b+GR1+ cells is not correct. Please double-check this important data. In addition, for the quantification of Ly6G, the authors should use Ly6G+ cells, not positive area.

Thank you for your comment.

After referring to the description of flow cytometry to detect neutrophils in the article of Huang et al.(Huang et al., 2013), we used CD11b^+^Ly6G^+^ to detect the number of neutrophils in mouse kidney tissue and peripheral blood. In Figure 4—figure supplement 1B and C, we mistakenly wrote Ly6G/Ly6C-FITC(GR1^+^), while we used FITC anti-mouse Ly-6G antibody (#127605, Biolegend). We first used APC anti mouse-CD45 antibody to detect leukocytes in mouse kidney tissue and peripheral blood by flow cytometry, then used PE anti-mouse / human CD11b antibody to detect myeloid cells in leukocytes, and finally detected the number of neutrophils by FITC anti-mouse Ly-6G antibody. We determined that there was an error in the quantification of Ly6G by careful examination, then we revised the quantification of Ly6G of Figure 4A, Figure 4F, Figure 4K and Figure 4—figure supplement 1D. And use Ly6G^+^ cells, not positive area.

References:

Huang H, Chen HW, Evankovich J, et al. Histones activate the NLRP3 inflammasome in Kupffer cells during sterile inflammatory liver injury. J Immunol. 2013;191(5):2665-2679. doi:10.4049/jimmunol.1202733.

5. It's difficult to believe that IL-19 promotes neutrophil infiltration as shown in Figure 7B. The authors should provide more solid data.

Thank you for your comment.

In Figure 9B, we mainly proved that IL-19 has chemotaxis on neutrophils. In order to further verify the role of IL-19 in promoting neutrophil infiltration in AAN mice, we injected IL-19 (0.2μg/g mice) into mice by intraperitoneal injection. After modeling, we sacrificed mice and detected the infiltration of neutrophils in mice's kidneys. IF (Immunofluorescence) staining showed that the number of MPO^+^ staining cells(neutrophils) in the kidney of mice. Compared with single intraperitoneal injection of AAI in mice, we found that IL-19 could aggravate the infiltration of neutrophils in the kidney of aristolochic acid nephropathy mice(Figure 8C).

6. Authors should check and review extensively for improvements to the use of English.

Thank you for your comment.

We used the Editage Language Editing service, revised the manuscript sentence by sentence and highlighted the paper.

Reviewer #2 (Recommendations for the authors):This study by Du et. al addressed the role and regulation of proline-serine-threonine phosphatase interacting protein 2 (PSTPIP2) and neutrophil extracellular traps (NETs) in Aristolochic acid Nephropthathy (AAN) and immune defense. PSTPIP2 expression is downregulated in AAN. Conditional knock-in of PSTPIP2 in mouse kidneys inhibited cell apoptosis, reduced neutrophil infiltration, suppressed the production of inflammatory factors and NETs, and ameliorated renal dysfunction. Reducing the expression of PSTPIP2 to normal levels in knock-in mouse using shRNA promoted kidney injury. Using in vivo model, the role of PSTPIP2 in AAN injury and renal function, apoptosis, neutrophil infiltration and NET formation is established. Using in vitro models, a PSTPIP2/NFkB-mediated NET formation via IL-19-IL20-β Receptor pathway is shown to induce inflammation and apoptosis in AAN. The studies are well presented.1. The studies could be strengthened by replicating the in vitro mechanistic pathway in the mouse model.

Thank you for your comment.

To confirm whether IL-19 can accelerate aristolochic acid nephropathy and to identify the functional role of IL-19 in NET formation in mice intraperitoneally injected with AAI, recombinant mouse IL-19 (rIL-19) was administered to mice immediately after treatment with AAI. Histological analysis with H&E staining revealed that intraperitoneal injection of rIL-19 at a dose of 4 µg caused no damage in vehicle-treated mice but considerably accelerated renal damage when combined with AAI treatment (Figure 7A). Similarly, intraperitoneal injection of IL-19 significantly increased the levels of Cr and BUN in the serum of AAI-treated mice (Figure 7B and C). In addition, western blotting and PCR and IHC analysis showed that KIM-1 expression was significantly upregulated in IL-19-treated AAN mice compared with vehicle-treated AAN mice (Figure 7D-F). Furthermore, we found that rIL-19 significantly increased TUNEL^+^ cell numbers (Figure 7I) and significantly upregulated the activity of caspase-3 and level of cleaved caspase-3 following intraperitoneal injection of IL-19 in AAI-treated mice (Figure 7G and H). These results suggest that injection of IL-19 protein accelerates aristolochic acid nephropathy. Intraperitoneal injection with AAI resulted in increased numbers of MPO^+^ neutrophils (Figure 8C) in the injured kidney, as well as increased serum levels of TNF-α and MCP-1 in rIL-19-treated mice compared to vehicle-treated mice. (Figure 8A and B). Similarly, NET formation, as measured by serum levels of MPO, dsDNA, and nucleosomes, as well as by tissue levels of Cit-H3 in the injured kidney, significantly increased in rIL-19-treated AAN mice compared to vehicle-treated AAN mice (Figure 8D-H). These results also indicated that IL-19 could further drive the inflammatory response and NET formation in aristolochic acid nephropathy.

2. The abstract should include a concluding sentence.

Thank you for your comment.

We added a concluding sentence in the paragraph of abstract and line 47-50.

“Our findings indicated that PSTPIP2 plays a key role in acute AAN and a novel cell communication mechanism between tubular epithelial cells and neutrophils in AAN mice, which could be some potential therapeutic targets.”